# A systematic review of substance use and substance use disorder research in Kenya

**Florence Jaguga**[1]*, **Sarah Kanana Kiburi**[2], **Eunice Temet**[3], **Julius Barasa**[4], **Serah Karanja**[5], **Lizz Kinyua**[6], **Edith Kamaru Kwobah**[1]

1 Department of Mental Health, Moi Teaching & Referral Hospital, Eldoret, Kenya, 2 Department of Mental Health, Mbagathi Hospital, Nairobi, Kenya, 3 Department of Mental Health & Behavioral Sciences, Moi University School of Medicine, Eldoret, Kenya, 4 Population Health, Academic Model Providing Access to Healthcare, Eldoret, Kenya, 5 Department of Mental Health, Gilgil Sub-County Hospital, Gilgil, Kenya, 6 Intensive Care Unit, Aga Khan University Hospital, Nairobi, Kenya

* flokemboi@gmail.com

**Data Availability Statement:** All relevant data are within the paper and its Supporting information files.

## Abstract

### Objectives

The burden of substance use in Kenya is significant. The objective of this study was to systematically summarize existing literature on substance use in Kenya, identify research gaps, and provide directions for future research.

### Methods

This systematic review was conducted in line with the PRISMA guidelines. We conducted a search of 5 bibliographic databases (PubMed, PsychINFO, Web of Science, Cumulative Index of Nursing and Allied Professionals (CINAHL) and Cochrane Library) from inception until 20 August 2020. In addition, we searched all the volumes of the official journal of the National Authority for the Campaign Against Alcohol & Drug Abuse (the African Journal of Alcohol and Drug Abuse). The results of eligible studies have been summarized descriptively and organized by three broad categories including: studies evaluating the epidemiology of substance use, studies evaluating interventions and programs, and qualitative studies exploring various themes on substance use other than interventions. The quality of the included studies was assessed with the Quality Assessment Tool for Studies with Diverse Designs.

### Results

Of the 185 studies that were eligible for inclusion, 144 investigated the epidemiology of substance use, 23 qualitatively explored various substance use related themes, and 18 evaluated substance use interventions and programs. Key evidence gaps emerged. Few studies had explored the epidemiology of hallucinogen, prescription medication, ecstasy, injecting drug use, and emerging substance use. Vulnerable populations such as pregnant women, and persons with physical disability had been under-represented within the epidemiological and qualitative work. No intervention study had been conducted among children and adolescents. Most interventions had focused on alcohol to the exclusion of other prevalent

**Funding:** The author(s) received no specific funding for this work.

**Competing interests:** The authors have declared that no competing interests exist.

**Abbreviations:** ASI, Addiction Severity Index; ASSIST, Alcohol Smoking and Substance Involvement Screening Test; AUD, Alcohol Use Disorder; AUDIT, Alcohol Use Identification Test; AUDIT-C, Alcohol Use Identification Test–Concise; BAI, Beck Anxiety Inventory; BDI, Beck Depression Inventory; BHS, Behavioral Health Screen; BMI, Body Mass index; BSIS, Beck Suicidal Intent Scale; CAD, Coronary Artery Disease; CAGE, Cut, Annoyed, Guilty, Eye-opener; CIDI, Composite International Diagnostic Interview; CINAHL, Cumulative Index of Nursing and Allied Professionals; CRAFFT, Car, Relax, Alone, Forget, Friends, Trouble; DAST, Drug Abuse Screening Test; DSM-III, Diagnostic & Statistical Manual Third Edition; DSM-III R, Diagnostic & Statistical Manual Third Edition Revised; DSM-IV, Diagnostic & Statistical Manual Fourth Edition; DSM-V, Diagnostic & Statistical Manual Fifth Edition; DUSI-R, Drug Use Screening Inventory—Revised; FGD, Focus Group Discussion; FSW, Female Sex Workers; GSHS, Global School-based Health Survey; HCV, Hepatitis C Virus; HCW, Healthcare worker; HIC, High Income Country; HIV, Human Immunodeficiency Virus; ICD, International Classification of Disease; IDI, In-depth Interviews; IDP, Internally Displaced Persons; IPV, Intimate Partner Violence; KIIs, Key Informant Interviews; K-SADS, Kiddie-Schedule for Affective Disorders; LGBTQ, Lesbian, Gay, Bisexual, Transgender, Queer; LMIC, Low and Middle Income Country; MAST, Michigan Alcohol Screening Test; MI, Motivational Interviewing; MINI, Mini International Neuropsychiatric Interview; MMT, Methadone Maintenance Therapy; MPBI, Multiple Problem Behavior Inventory; MSM, Men who have Sex with Men; MSME, Men who have Sex with Men Exclusively; MSMW, Men who have Sex with Men & Women; NIH, National Institute of Health; NSP, Needle Syringe Program; OST, Opioid Substitution Therapy; PLHIV, People Living with HIV; PrEP, Pre-exposure Prophylaxis; PTSD, Post-Traumatic Stress Disorder; PWID, People Who Inject Drugs; QATSDD, Quality Assessment Tool for Studies with Diverse Designs; RCT, Randomized controlled trial; RTAs, Road Traffic Accidents; SCID, Structured Clinical interview for DSM; SES, Socio-economic Status; SSA, Sub-Saharan Africa; TB, Tuberculosis; UNODC, United Nations Office on Drugs and Crime; VCT, Voluntary Counseling & Testing; WOTC, Wisdom of the Crowds.

substances such as tobacco and cannabis. Little had been done to evaluate digital and population-level interventions.

## Conclusion

The results of this systematic review provide important directions for future substance use research in Kenya.

## Systematic review registration

PROSPERO: CRD42020203717.

## Introduction

Globally, substance use is associated with significant morbidity and mortality. In the 2017 Global Burden of Disease (GBD) study, substance use disorders (SUDs) were the second leading cause of disability among the mental disorders with 31,052,000 (25%) Years Lived with Disability (YLD) attributed to them [1]. In 2016, harmful alcohol use resulted in 3 million deaths (5.3% of all deaths) worldwide and 132.6 (5.1%) million disability-adjusted life years (DALYs) [2]. Tobacco use, the leading cause of preventable death, kills more than 8 million people worldwide annually [3]. Alcohol and tobacco use are leading risk factors for non-communicable diseases for example cardiovascular disease, cancer, and liver disease [3, 4]. Even though the prevalence rate of opioid use is small compared to that of tobacco and alcohol use, opioid use disorder contributes to 76% of all deaths from SUDs [4]. Other psychoactive substances such as cannabis and amphetamines are associated with mental health consequences including increased risk of suicidality, depression, anxiety and psychosis [5, 6]. In addition to the effect on health, substance use is associated with significant socio-economic costs arising from its impact on health and criminal justice systems [7].

Low- and middle-income countries (LMICs) bear the burden of substance use. Over 80% of the 1.3 billion tobacco users worldwide live in LMICs [3]. In 2016, the alcohol-attributable disease burden was highest in LMICs compared to upper-middle-income and high-income countries (HICs) [2]. In Kenya, a nationwide survey conducted in 2017 reported that over 10% of Kenyans between the ages of 15 to 65 years had a SUD [8]. In another survey, 20% of primary school children had ever used at least one substance in their lifetime [9]. Moreover, Kenya has the third highest total DALYs (54,000) from alcohol use disorders (AUD) in Africa [4] Unfortunately, empirical work on substance use in LMICs is limited [10, 11]. In a global mapping of SUD research, majority of the work had been conducted in upper-middle income and HICs (HICs) [11]. In a study whose aim was to document the existing work on mental health in Botswana, only 7 studies had focused on substance use [10]. Information upon which policy and interventions could be developed is therefore lacking in low-and-middle income settings.

Since the early 1980s, scholars in Kenya began engaging in research to document the burden and patterns of substance use [12]. In 2001 the National Authority for the Campaign Against Alcohol and Drug Abuse (NACADA) was established in response to the rising cases of harmful substance use in the country particularly among the youth. The mandate of the Authority was to educate the public on the harms associated with substance use [13]. In addition to prevention work, NACADA contributes to research by conducting general population prevalence surveys every 5 years and recently launched its journal, the African Journal of

Alcohol and Drug Abuse (AJADA) [14]. The amount of empirical work done on substance use in Kenya has expanded since these early years but has not been systematically summarized. The evidence gaps therefore remain unclear.

In order to guide future research efforts and adequately address the substance use scourge in Kenya, there is need to document the scope and breadth of available scientific literature. The aim of this systematic review is therefore: (i) to describe the characteristics of research studies conducted on substance use and SUD in Kenya; (ii) to assess the methodological quality of the studies; (iii) to identify areas where there is limited research evidence and; (iv) to make recommendations for future research. This paper is in line the Vision 2030 [15], Kenya's national development policy framework, which directs that the government implements substance use treatment and prevention projects and programs, and target 3.5 of the Sustainable Development Goals (SDGs) which requires that countries strengthen the treatment and prevention for SUDs [16].

## Materials and methods

### Protocol and registration

In conducting this systematic review we adhered to the recommendations from the Preferred Reporting Items for Systematic Reviews and Meta-Analyses (PRISMA) statement [17]. A 27-item PRISMA checklist is available as an additional file to this protocol (S1 Checklist). Our protocol was registered in the International Prospective Register of Systematic Reviews (PROSPERO): CRD42020203717.

### Search strategy

A search was carried out in five electronic databases on 20th August 2020: PubMed, PsychINFO, Web of Science, Cumulative Index of Nursing and Allied Professionals (CINAHL) and Cochrane Library. The full search strategy can be found in S1 File and takes the following form: *(terms for substance use) and (terms for substance use outcomes of interest) and (terms for region)*. The searches spanned the period from inception to date. No filter was applied. A manual search was done in Volumes 1, 2 and 3 (all published volumes by the time of the search) of the recently launched AJADA journal by NACADA, and additional articles identified. [14, 18, 19].

### Study selection

Following the initial search, all articles were loaded onto Mendeley reference manager where initial duplicate screening and removal was done. After duplicate removal, the articles were loaded onto Rayyan, a soft-ware for screening and selecting studies during the conduct of systematic reviews [20]. The abstract and titles of retrieved articles were independently screened by two authors based on a set of pre-determined eligibility criteria. A second screening of full text articles was also done independently by two authors and resulted in an 88.7% agreement. Disagreements during each stage of the screening were resolved through discussion and consensus.

### Inclusion criteria

Since we sought to map existing literature on the subject, our inclusion criteria were broad. We included articles on substance use if (i) the sample or part of the sample was from Kenya, (ii) they were original research articles, (iii) they had a substance use or SUD exposure, (iv) they had a substance use or SUD related outcome such as prevalence, pattern of use, prevention and treatment, and (iv) they were published in English or had an English translation

available. We included studies conducted among all age groups and studies that used all designs including quantitative, qualitative and mixed methods.

## Exclusion criteria

Studies were excluded if: (i) they were cross-national and did not report country specific results (ii) they did not report substance use or SUD as an exposure, and did not have substance use or SUD related outcomes or as part of the outcomes, (iii) they were review articles, dissertations, conference presentations or abstracts, commentaries or editorials, (iv) and the full text articles were not available.

## Data extraction

We prepared 3 data extraction forms based on three emerging categories of studies i.e.:

- Studies reporting on the epidemiology of substance use or SUD

- Studies evaluating substance use or SUD interventions and programs

- Studies qualitatively exploring various themes on substance use or SUD (but not evaluating interventions or programs)

The forms were piloted by F.J. and S.K. and adjustments made to the content. Data extraction was then done using the final form by all authors and double checked by F.J. for completeness and accuracy. Discrepancies were resolved by discussion with S.K. and E.T. until consensus was achieved. The following data was extracted for each study category:

1. Studies reporting on the epidemiology of substance use or SUD: study design, study population characteristics, study setting, sample size, age and gender distribution, substance(s) assessed, standardized tool or criteria used, main findings (prevalence, risk factors, other key findings).

2. Studies evaluating substance use or SUD interventions and programs: study design, study objective, sample size, name of the intervention or program, person delivering intervention, outcomes and measures, and main findings.

3. Studies qualitatively exploring various aspects of substance use or SUD other than programs and interventions: study objective, methods of data collection, study setting, study population, age and gender distribution, theoretical framework used, and main findings.

## Data synthesis

The results have been summarized descriptively and organized by the three categories above. Within each category, a general description of the study characteristics has been provided followed by a narrative synthesis of findings organized by sub-themes inductively derived from the data. The sub-themes within each category are as follows:

**Studies reporting on the epidemiology of substance use or SUD**: Epidemiology of alcohol use, epidemiology of tobacco use, epidemiology of khat use, epidemiology of cannabis use, epidemiology of opioid and cocaine use, epidemiology of other substance use (sedatives, inhalants, hallucinogens, prescription medication, emerging drugs, ecstasy).

**Studies evaluating substance use or SUD interventions and programs:** *Individual level interventions* (Individual-level interventions for harmful alcohol use, individual-level interventions for khat use, individual level intervention for substance use in general); ***Programs***

(Methadone programs, needle-syringe programs, tobacco cessation programs, out-patient SUD treatment programs); ***Population-level interventions***: Population-level tobacco interventions, population-level alcohol interventions.

**Studies qualitatively exploring various aspects of substance use or SUD other than programs and interventions**: Injecting drug use and heroin use, alcohol use, substance use among youth and adolescents, other topics.

## Quality assessment of the studies

Quality assessment was conducted by S.K. using the Quality Assessment Tool for Studies with Diverse Designs (QATSDD) [21]. F.J. & J.B. double checked the scores for completeness and accuracy. Any disagreements were discussed and resolved by consensus. We had initially planned to use the National Institute of Health (NIH) set of quality assessment tools but due to the diverse nature of study designs, the authors agreed to use the QATSDD tool. The QATSDD is a 16-item tool for both qualitative and quantitative studies. Each item is scored on a 4-point scale (0–3), with a total of 14 criteria for each study design and 16 for studies with mixed methods. Scoring relies on guidance notes provided as well as judgment and expertise from the reviewers. The criteria used are: (i) theoretical framework; (ii) statement of aims or objectives; (iii) description of research setting; (iv) sample size consideration; (v) representative sample of target group (vi) data collection procedure description; (vii) rationale for choice of data collection tool(s); (viii) detailed recruitment data; (ix) statistical assessment of reliability and validity of measurement tools (quantitative only); (x) fit between research question and method of data collection (quantitative only); (xi) fit between research question and format and content data collection (qualitative only); (xii) fit between research question and method of analysis; (xiii) justification of analytical method; (xiv) assessment of reliability of analytical process (qualitative only); (xv) user involvement in design and (xvi) discussion on strengths and limitations[21]. Scores are awarded for each criterion as follows: 0 = no mention at all; 1 = very brief description; 2 = moderate description; and 3 = complete description. The scores of each criterion are then summed up with a maximum score of 48 for mixed methods studies and 42 for studies using either qualitative only or quantitative only designs. For ease of interpretation, the scores were converted to percentages and classified as low (<50%), medium (50%–80%) or high (>80%) quality of evidence [22].

## Results

### Search results

The search from the five electronic databases yielded 1535 results: 950 from PubMed, 173 from PsychINFO, 210 from web of science, 123 from CINAHL and 79 from Cochrane library. Thirteen additional studies were identified through a manual search of the AJADA journals (Volumes 1, 2 and 3). Studies were assessed for duplicates and 1154 articles remained after removal of duplicates. The 1154 studies underwent an initial screening based on abstracts and titles, and 946 articles were excluded. A second screen of full text articles was done for the 208 studies that were potentially eligible for the review. Twenty three studies were excluded as follows: 21 did not meet the eligibility criteria and 2 had duplicated results. A total of 185 studies were found to meet the inclusion criteria and were included in the review (Fig 1).

### General characteristics of the studies

Of the 185 studies included in this review, 144 (77.8%) investigated the epidemiology of substance use or SUD, 18 (9.7%) evaluated substance use or SUD interventions and programs,

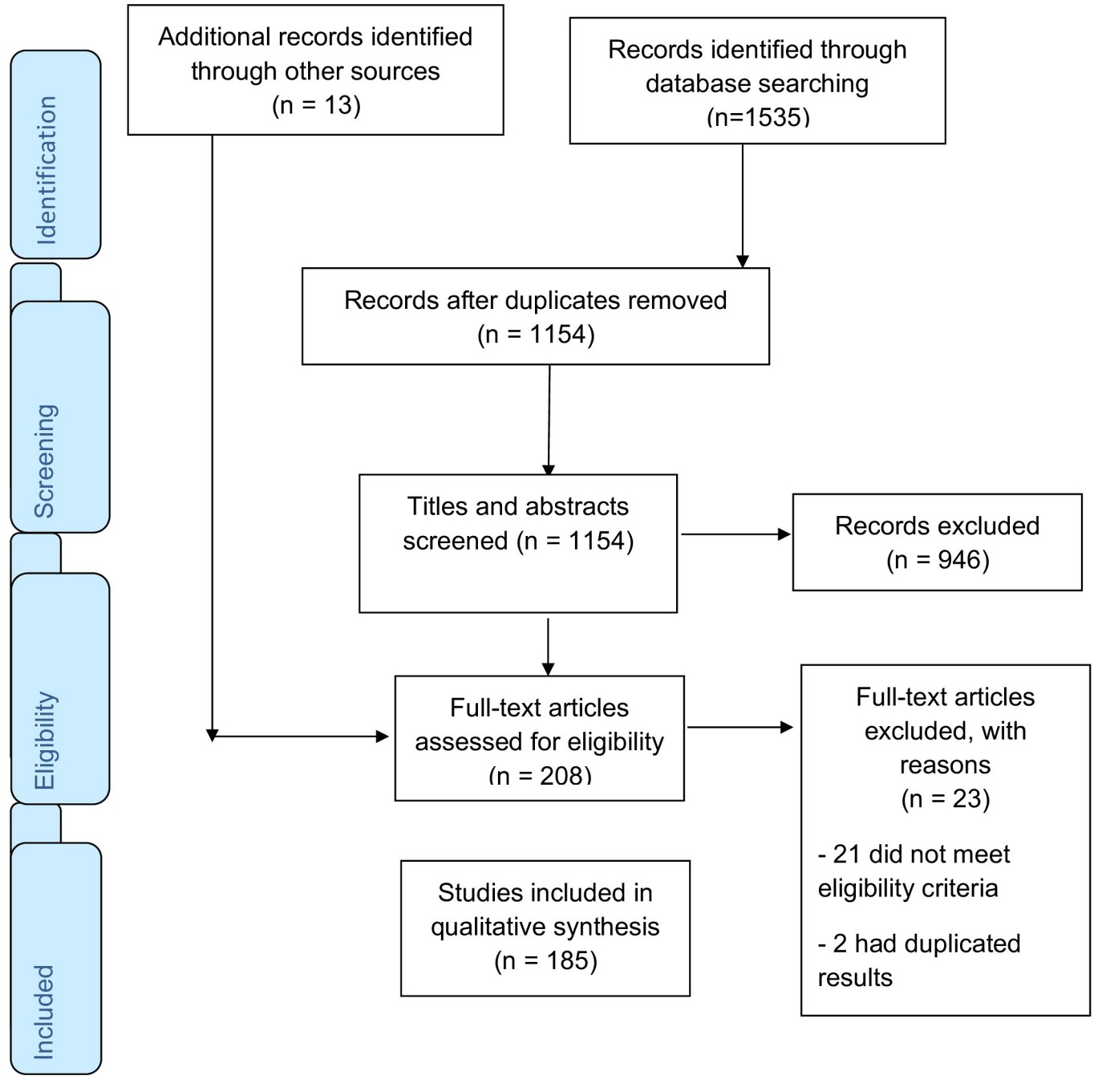

**Fig 1. PRISMA flow chart.**

and 23 (12.4%) were qualitative studies exploring perceptions on various substance use or SUD topics other than interventions and programs (Table 4). The studies were published between 1982 and 2020. The number of studies published has gradually increased in number over the years, particularly in the past decade. Fig 2 shows the publication trends for substance use research in Kenya.

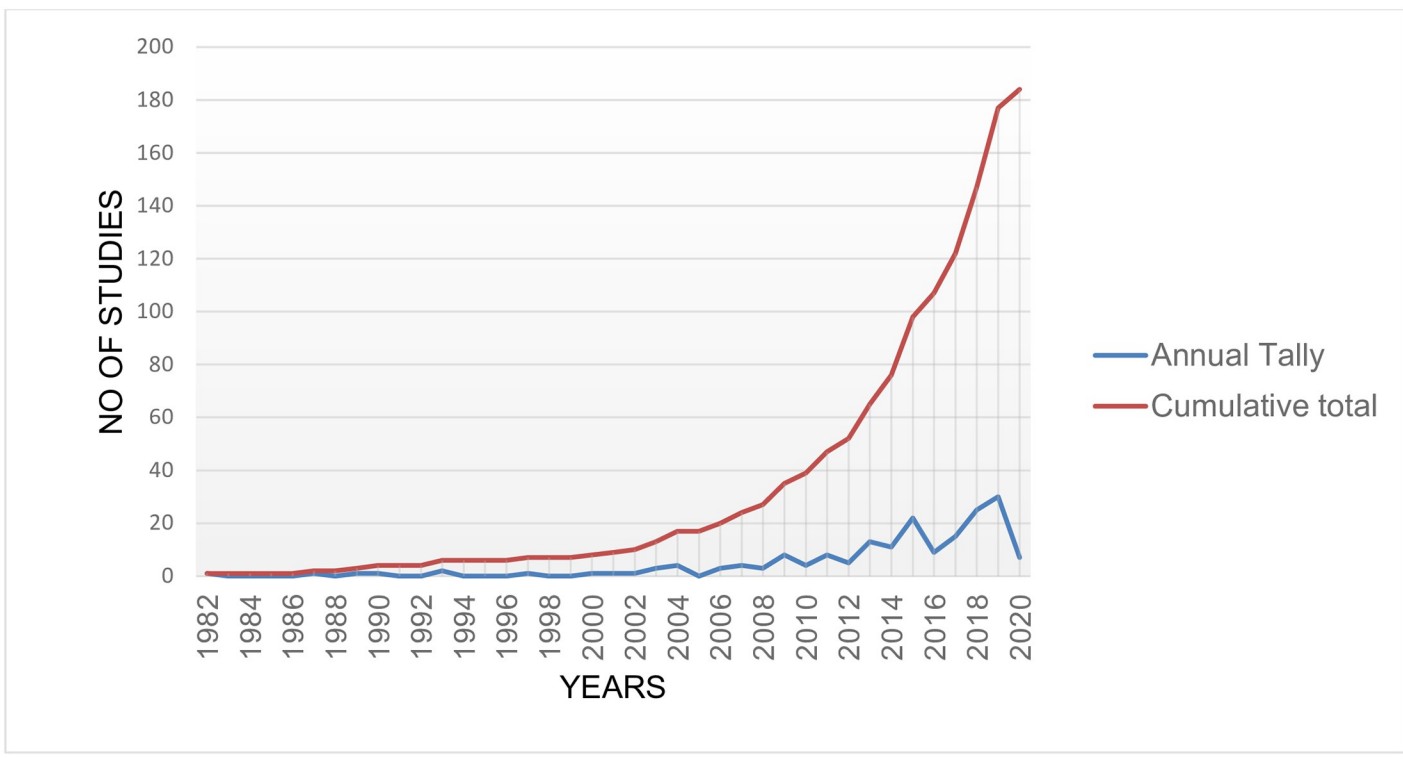

**Fig 2. Line graph showing publication trends for substance use research in Kenya.**

## Quality assessment

The QATSDD scores ranged from 28.6% [23] to 92.9% [24]. Only 14 studies [12, 23, 25–36] (all quantitative) had scores of less than 50%. Of these, the main items driving low quality were: no mention of user involvement in study design (n = 14) [12, 23, 25–36], no explicit mention of a theoretical framework (n = 10) [12, 23, 25–28, 30, 33, 35, 36] and a lack of a statistical assessment of reliability and validity of measurement tools (n = 10) [12, 23, 25, 28, 30–33, 35, 36] Table 1.

## Studies examining the epidemiology of substance use or SUD

**General description of epidemiological studies.** One hundred and forty-four studies examined the prevalence and or risk factors for various substances. The studies were published between 1982 and 2020. The four main study designs used were cross-sectional (n = 126), cohort (n = 5), case-control (n = 10), and mixed methods (n = 2). One study used a combination of the multiplier method, Wisdom of the Crowds (WOTC) method, and a published literature review to document the size of key populations [164]. The sample size for this category of studies ranged from 42 [130] to 72292 [128].

The studies were conducted in diverse settings including the community (n = 72), hospitals (n = 40), institutions of learning (n = 24), streets (n = 5), prisons and courts (n = 3), charitable institutions (n = 1), methadone maintenance therapy (MMT) clinics (n = 1), and in needle-syringe program (NSP) sites (n = 1). Of the studies conducted within the community, 12 were conducted in informal settlements. The study populations were similarly diverse as follows: general population adults & adolescents (n = 39), persons with NCDs (n = 11), primary and secondary school students (n = 15), people who inject drugs (PWID) (n = 11), general patients

**Table 1. Quality assessment.**

| Mixed Methods studies | | | | | | | | | | | | | | | | | | |
|---|---|---|---|---|---|---|---|---|---|---|---|---|---|---|---|---|---|---|
| Author, year | 1 | 2 | 3 | 4 | 5 | 6 | 7 | 8 | 9 | 10 | 11 | 12 | 13 | 14 | 15 | 16 | Total/48 | Percentage of total |
| Kamenderi et al., 2020 [37] | 1 | 3 | 3 | 3 | 3 | 3 | 2 | 3 | 2 | 3 | 3 | 3 | 2 | 0 | 0 | 1 | 35 | 72.9 |
| Mackenzie et al., 2009 [38] | 1 | 3 | 1 | 3 | 3 | 3 | 3 | 3 | 2 | 3 | 3 | 3 | 3 | 0 | 0 | 0 | 34 | 70.8 |
| Mutai et al., 2020 [39] | 3 | 3 | 3 | 2 | 3 | 3 | 2 | 3 | 1 | 3 | 3 | 3 | 1 | 0 | 0 | 1 | 34 | 70.8 |
| Papas et al., 2010 [40] | 2 | 3 | 3 | 2 | 3 | 3 | 3 | 3 | 2 | 3 | 3 | 3 | 3 | 3 | 1 | 2 | 42 | 87.5 |
| **Qualitative studies** | | | | | | | | | | | | | | | | | | |
| Author, Year | 1 | 2 | 3 | 4 | 5 | 6 | 7 | 8 | 9 | 10 | 11 | 12 | 13 | 14 | 15 | 16 | Total/42 | Percentage of total |
| Bazzi et al., 2019 [41] | 3 | 3 | 2 | 1 | 2 | 2 | 1 | 2 | NA | NA | 3 | 2 | 2 | 3 | 0 | 2 | 28 | 66.7 |
| Beckerleg 2004 [42] | 0 | 2 | 3 | 3 | 2 | 1 | 2 | 2 | NA | NA | 3 | 3 | 1 | 1 | 0 | 0 | 23 | 54.8 |
| Ezard et al., 2011 [43] | 2 | 3 | 2 | 2 | 3 | 3 | 2 | 3 | NA | NA | 3 | 3 | 3 | 3 | 1 | 0 | 33 | 78.6 |
| Guise et al., 2015 [44] | 2 | 3 | 3 | 3 | 3 | 3 | 3 | 3 | NA | NA | 3 | 3 | 3 | 3 | 0 | 2 | 37 | 88.1 |
| Guise et al., 2019 [45] | 3 | 3 | 2 | 2 | 2 | 2 | 1 | 2 | NA | NA | 3 | 3 | 2 | 3 | 0 | 1 | 29 | 69.0 |
| Kibicho & Campbell, 2019 [46] | 2 | 3 | 3 | 3 | 2 | 3 | 3 | 3 | NA | NA | 3 | 3 | 2 | 2 | 0 | 3 | 35 | 83.3 |
| Mburu et al., 2018 [47] | 1 | 3 | 3 | 2 | 3 | 2 | 3 | 3 | NA | NA | 3 | 3 | 2 | 3 | 0 | 2 | 33 | 78.6 |
| Mburu et al., 2019 [48] | 3 | 2 | 3 | 3 | 3 | 2 | 2 | 2 | NA | NA | 3 | 3 | 3 | 3 | 0 | 3 | 35 | 83.3 |
| Mburu et al., 2020 [49] | 1 | 2 | 3 | 3 | 3 | 3 | 2 | 3 | NA | NA | 3 | 3 | 3 | 3 | 0 | 2 | 34 | 81.0 |
| Mburu et al., 2019 [50] | 3 | 3 | 3 | 3 | 3 | 3 | 2 | 3 | NA | NA | 3 | NA | 3 | 3 | 0 | 3 | 35 | 83.3 |
| Mburu, 2018 [51] | 3 | 3 | 3 | 3 | 3 | 3 | 3 | 3 | NA | NA | 3 | NA | 3 | 3 | 0 | 3 | 36 | 85.7 |
| Mital et al., 2016 [52] | 1 | 3 | 3 | 3 | 2 | 3 | 2 | 2 | NA | NA | 3 | 3 | 3 | 3 | 0 | 2 | 33 | 78.6 |
| Muturi, 2014 [53] | 2 | 3 | 1 | 3 | 3 | 2 | 2 | 2 | NA | NA | 3 | 3 | 3 | 3 | 0 | 3 | 33 | 78.6 |
| Muturi, 2015 [54] | 3 | 3 | 1 | 3 | 3 | 2 | 2 | 2 | NA | NA | 3 | 3 | 3 | 3 | 0 | 0 | 31 | 73.8 |
| Muturi et al., 2016 [55] | 3 | 3 | 1 | 3 | 3 | 2 | 2 | 2 | NA | NA | 3 | 3 | 3 | 3 | 0 | 2 | 33 | 78.6 |
| Ndimbii et al., 2015 [56] | 1 | 1 | 2 | 3 | 3 | 2 | 1 | 2 | NA | NA | 3 | 3 | 3 | 3 | 0 | 0 | 27 | 64.3 |
| Ndimbii et al., 2018 [57] | 1 | 3 | 2 | 3 | 3 | 3 | 3 | 3 | NA | NA | 3 | 3 | 2 | 3 | 2 | 3 | 37 | 88.1 |
| Njue et al., 2009 [58] | 0 | 2 | 2 | 3 | 2 | 2 | 1 | 2 | NA | NA | 3 | 3 | 2 | 2 | 2 | 2 | 28 | 66.7 |
| Njue et al., 2011 [59] | 0 | 3 | 2 | 3 | 3 | 3 | 2 | 2 | NA | NA | 3 | 3 | 2 | 2 | 2 | 0 | 30 | 71.4 |
| Othieno et al., 2012 [60] | 2 | 2 | 3 | 2 | 2 | 3 | 1 | 2 | NA | NA | 3 | 3 | 1 | 2 | 0 | 2 | 27 | 64.3 |
| Rhodes et al., 2015 [61] | 3 | 2 | 3 | 3 | 3 | 3 | 1 | 3 | NA | NA | 3 | 3 | 3 | 3 | 0 | 0 | 33 | 78.6 |
| Rhodes, 2018 [62] | 2 | 2 | 3 | 3 | 3 | 2 | 1 | 2 | NA | NA | 3 | 3 | 3 | 3 | 0 | 0 | 30 | 71.4 |
| Ssewanyana et al., 2018 [63] | 3 | 3 | 3 | 3 | 3 | 3 | 2 | 3 | NA | NA | 3 | 3 | 3 | 3 | 0 | 3 | 38 | 90.5 |
| Syvertsen et al., 2016 [64] | 0 | 2 | 2 | 2 | 3 | 1 | 0 | 1 | NA | NA | 3 | 3 | 2 | 2 | 0 | 1 | 22 | 52.3 |
| Syvertsen et al., 2019 [65] | 0 | 2 | 1 | 1 | 3 | 1 | 1 | 1 | NA | NA | 3 | 3 | 3 | 3 | 0 | 1 | 23 | 54.8 |
| Velloza et al., 2015 [66] | 3 | 3 | 3 | 3 | 3 | 3 | 2 | 3 | NA | NA | 3 | 3 | 3 | 3 | 0 | 3 | 38 | 90.5 |
| Yotebieng et al., 2016 [67] | 1 | 3 | 3 | 3 | 3 | 3 | 2 | 3 | NA | NA | 3 | 3 | 3 | 3 | 0 | 2 | 35 | 83.3 |
| **Quantitative studies** | | | | | | | | | | | | | | | | | | |
| First author, Year | 1 | 2 | 3 | 4 | 5 | 6 | 7 | 8 | 9 | 10 | 11 | 12 | 13 | 14 | 15 | 16 | Total/42 | Percentage of total |
| Aden et al., 2006 [23] | 0 | 3 | 3 | 0 | 1 | 2 | 0 | 3 | 0 | 0 | NA | 0 | 0 | NA | 0 | 0 | 12 | 28.6 |
| Akiyama et al., 2019 [68] | 1 | 3 | 2 | 3 | 3 | 2 | 1 | 2 | 1 | 3 | NA | 3 | 3 | NA | 0 | 3 | 30 | 71.4 |
| Anundo, 2019 [69] | 3 | 3 | 3 | 2 | 2 | 3 | 3 | 2 | 2 | 3 | NA | 3 | 1 | NA | 0 | 0 | 30 | 71.4 |
| Asiki et al., 2018 [70] | 3 | 3 | 3 | 1 | 3 | 3 | 3 | 3 | 3 | 3 | NA | 3 | 3 | NA | 2 | 2 | 38 | 90.5 |
| Astrom et al., 2004 [71] | 0 | 3 | 3 | 3 | 3 | 2 | 3 | 1 | 1 | 3 | NA | 2 | 0 | NA | 0 | 0 | 24 | 57.1 |
| Atwoli et al., 2011 [72] | 1 | 3 | 3 | 3 | 3 | 3 | 3 | 3 | 1 | 3 | NA | 3 | 2 | NA | 0 | 2 | 33 | 78.6 |
| Ayah et al., 2013 [73] | 1 | 3 | 3 | 3 | 3 | 3 | 3 | 3 | 1 | 3 | NA | 3 | 2 | NA | 0 | 3 | 34 | 81.0 |
| Ayaya et al., 2002 [74] | 1 | 3 | 3 | 2 | 2 | 3 | 2 | 3 | 1 | 3 | NA | 3 | 2 | NA | 0 | 2 | 30 | 71.4 |
| Balogun et al., 2014 [75] | 2 | 3 | 2 | 1 | 3 | 1 | 1 | 1 | 1 | 3 | NA | 3 | 3 | NA | 0 | 3 | 27 | 64.3 |
| Beckerlerg et al., 2006 [76] | 0 | 3 | 3 | 2 | 3 | 3 | 1 | 3 | 2 | 3 | NA | 3 | 1 | NA | 0 | 0 | 27 | 64.3 |
| Bengston et al., 2014 [77] | 1 | 3 | 3 | 2 | 2 | 2 | 3 | 2 | 2 | 3 | NA | 3 | 2 | NA | 0 | 3 | 31 | 73.8 |

*(Continued)*

**Table 1.** (*Continued*)

| | | | | | | | | | | | | | | | | | | |
|---|---|---|---|---|---|---|---|---|---|---|---|---|---|---|---|---|---|---|
| Budambula et al., 2018 [78] | 1 | 3 | 1 | 3 | 3 | 3 | 2 | 2 | 2 | 3 | NA | 3 | 3 | NA | 3 | 3 | 35 | 83.3 |
| Cagle et al., 2018 [79] | 1 | 3 | 3 | 0 | 2 | 2 | 2 | 2 | 2 | 3 | NA | 2 | 2 | NA | 0 | 2 | 26 | 61.9 |
| Chersich et al., 2014 [80] | 3 | 3 | 3 | 2 | 2 | 3 | 3 | 3 | 2 | 3 | NA | 3 | 3 | NA | 0 | 3 | 36 | 85.7 |
| Christensen et al., 2009 [81] | 0 | 3 | 3 | 2 | 3 | 3 | 3 | 3 | 2 | 3 | NA | 3 | 3 | NA | 0 | 0 | 31 | 73.8 |
| Cleland et al. 2007 [82] | 0 | 3 | 3 | 0 | 0 | 3 | 3 | 3 | 0 | 3 | NA | 3 | 3 | NA | 0 | 3 | 27 | 64.2 |
| De Menil et al., 2014 [83] | 1 | 3 | 3 | 1 | 2 | 2 | 2 | 1 | 1 | 3 | NA | 2 | 1 | NA | 0 | 2 | 24 | 57.1 |
| Deveau Dhadphale, 2010 [84] | 1 | 2 | 3 | 2 | 3 | 2 | 1 | 3 | 1 | 3 | NA | 3 | 1 | NA | 0 | 0 | 25 | 59.5 |
| Dhadphale et al., 1982 [12] | 0 | 2 | 1 | 1 | 2 | 1 | 0 | 1 | 0 | 3 | NA | 3 | 0 | NA | 0 | 0 | 14 | 33.3 |
| Dhadphale, 1997 [27] | 0 | 3 | 1 | 1 | 2 | 1 | 1 | 1 | 2 | 3 | NA | 3 | 0 | NA | 0 | 0 | 18 | 42.9 |
| Embleton, 2012 [85] | 3 | 3 | 3 | 3 | 2 | 3 | 3 | 3 | 2 | 3 | NA | 3 | 3 | NA | 0 | 3 | 37 | 88.1 |
| Embleton et al., 2013[86] | 2 | 3 | 3 | 3 | 2 | 3 | 3 | 3 | 2 | 3 | NA | 3 | 3 | NA | 0 | 3 | 36 | 85.7 |
| Embleton et al., 2017 [87] | 1 | 3 | 3 | 3 | 2 | 3 | 3 | 3 | 2 | 3 | NA | 3 | 3 | NA | 0 | 3 | 35 | 83.3 |
| Gathecha et al., 2018 [88] | 2 | 3 | 2 | 2 | 3 | 3 | 3 | 3 | 2 | 3 | NA | 3 | 3 | NA | 0 | 3 | 35 | 83.3 |
| Gichuki et al., 2015 [89] | 3 | 3 | 3 | 3 | 2 | 3 | 3 | 3 | 3 | 3 | NA | 3 | 2 | NA | 1 | 2 | 37 | 88.1 |
| Gitatui et al., 2019 [24] | 2 | 3 | 3 | 3 | 2 | 3 | 3 | 3 | 3 | 3 | NA | 3 | 3 | NA | 3 | 2 | 39 | 92.9 |
| Giusto et al., 2020 [90] | 3 | 3 | 3 | 2 | 3 | 3 | 3 | 2 | 3 | 3 | NA | 3 | 3 | NA | 1 | 3 | 38 | 90.5 |
| Goldblatt et al., 2015 [91] | 2 | 3 | 3 | 2 | 2 | 3 | 3 | 3 | 3 | 3 | NA | 3 | 3 | NA | 1 | 3 | 37 | 88.1 |
| Goodman et al., 2017 [92] | 2 | 3 | 3 | 3 | 3 | 3 | 3 | 3 | 2 | 3 | NA | 3 | 3 | NA | 0 | 3 | 37 | 88.1 |
| Hall et al., 1993 [93] | 1 | 2 | 3 | 2 | 3 | 2 | 3 | 2 | 2 | 3 | NA | 3 | 3 | NA | 0 | 0 | 29 | 69.0 |
| Harder et al., 2019 [94] | 3 | 3 | 3 | 3 | 3 | 3 | 3 | 3 | 2 | 3 | NA | 3 | 3 | NA | 0 | 3 | 38 | 90.5 |
| Haregu et al., 2019 [95] | 1 | 3 | 3 | 3 | 3 | 3 | 2 | 3 | 2 | 3 | NA | 3 | 3 | NA | 0 | 2 | 34 | 81.0 |
| Hulzebosch et al., 2015 [96] | 1 | 3 | 3 | 2 | 3 | 3 | 2 | 3 | 2 | 3 | NA | 3 | 3 | NA | 0 | 3 | 34 | 81.0 |
| Jenkins et al., 2017 [97] | 1 | 3 | 3 | 2 | 2 | 3 | 3 | 3 | 2 | 3 | NA | 3 | 3 | NA | 0 | 2 | 33 | 78.6 |
| Joshi et al., 2015 [98] | 1 | 2 | 3 | 3 | 2 | 3 | 3 | 3 | 2 | 3 | NA | 3 | 3 | NA | 0 | 3 | 34 | 81.0 |
| Kaai et al., 2019 [99] | 1 | 2 | 2 | 2 | 3 | 2 | 2 | 2 | 1 | 3 | NA | 3 | 3 | NA | 0 | 2 | 28 | 66.7 |
| Kaduka et al., 2017 [100] | 0 | 3 | 3 | 3 | 3 | 2 | 2 | 1 | 2 | 3 | NA | 3 | 2 | NA | 0 | 0 | 27 | 64.3 |
| Kamau et al., 2017 [101] | 0 | 3 | 3 | 2 | 2 | 2 | 3 | 3 | 2 | 3 | NA | 3 | 3 | NA | 0 | 2 | 30 | 71.4 |
| Kamenderi, 2019 [102] | 1 | 3 | 2 | 3 | 3 | 1 | 1 | 2 | 1 | 3 | NA | 3 | 3 | NA | 0 | 0 | 26 | 61.9 |
| Kamenderi et al., 2019 [103] | 1 | 3 | 3 | 3 | 3 | 3 | 1 | 3 | 1 | 3 | NA | 3 | 2 | NA | 0 | 0 | 29 | 69.0 |
| Kamenderi et al., 2019 [104] | 1 | 3 | 3 | 3 | 3 | 3 | 2 | 3 | 2 | 3 | NA | 3 | 2 | NA | 0 | 1 | 32 | 76.2 |
| Kamotho et al., 2004[105] | 1 | 3 | 2 | 2 | 2 | 2 | 2 | 2 | 1 | 2 | NA | 3 | 2 | NA | 0 | 2 | 26 | 61.9 |
| Kanyanya et al., 2007 [31] | 2 | 3 | 3 | 1 | 3 | 1 | 0 | 1 | 0 | 3 | NA | 3 | 0 | NA | 0 | 0 | 20 | 47.6 |
| Kaplan et al., 1990 [106] | 1 | 2 | 3 | 2 | 3 | 2 | 1 | 2 | 0 | 3 | NA | 3 | 0 | NA | 0 | 1 | 23 | 54.8 |
| Kendagor et al., 2018 [107] | 2 | 3 | 3 | 2 | 3 | 2 | 3 | 2 | 2 | 3 | NA | 3 | 3 | NA | 0 | 2 | 33 | 78.6 |
| Khasakhala et al., 2013 [108] | 1 | 3 | 3 | 3 | 2 | 3 | 3 | 3 | 2 | 3 | NA | 3 | 3 | NA | 0 | 2 | 34 | 81.0 |
| Khasakhala et al., 2013 [109] | 1 | 2 | 3 | 3 | 2 | 3 | 3 | 3 | 2 | 3 | NA | 3 | 3 | NA | 0 | 2 | 33 | 78.6 |
| Kiburi et al., 2018 [110] | 1 | 3 | 3 | 2 | 2 | 2 | 3 | 2 | 2 | 3 | NA | 3 | 3 | NA | 0 | 2 | 31 | 73.8 |
| Kimando et al., 2017 [111] | 1 | 2 | 3 | 2 | 2 | 2 | 2 | 2 | 1 | 3 | NA | 3 | 3 | NA | 0 | 0 | 26 | 61.9 |
| Kimani et al., 2019 [112] | 1 | 3 | 3 | 3 | 2 | 2 | 1 | 2 | 0 | 3 | NA | 3 | 3 | NA | 0 | 2 | 28 | 66.7 |
| Kimbui et al., 2018[113] | 3 | 3 | 3 | 2 | 3 | 3 | 3 | 3 | 2 | 3 | NA | 3 | 3 | NA | 0 | 2 | 36 | 85.7 |
| Kinoti et al., 2011 [114] | 1 | 3 | 3 | 1 | 2 | 2 | 2 | 2 | 1 | 3 | NA | 3 | 0 | NA | 0 | 2 | 25 | 59.5 |
| Kinyanjui & Atwoli, 2013 [115] | 0 | 3 | 3 | 2 | 3 | 3 | 3 | 3 | 2 | 3 | NA | 3 | 2 | NA | 0 | 2 | 32 | 76.2 |
| Kisilu et al., 2019 [29] | 1 | 3 | 1 | 2 | 2 | 1 | 1 | 1 | 1 | 2 | NA | 3 | 1 | NA | 0 | 0 | 19 | 45.2 |
| Komu et al., 2009[116] | 0 | 3 | 2 | 2 | 2 | 1 | 1 | 1 | 0 | 3 | NA | 3 | 1 | NA | 0 | 2 | 21 | 50.0 |
| Korhonen et al., 2018 [117] | 1 | 2 | 3 | 1 | 3 | 3 | 3 | 3 | 2 | 3 | NA | 3 | 3 | NA | 0 | 3 | 33 | 78.6 |
| Kunzweiler et al., 2017 [118] | 0 | 3 | 3 | 2 | 0 | 3 | 0 | 3 | 0 | 3 | NA | 3 | 3 | NA | 0 | 3 | 26 | 61.9 |
| Kunzweiler et al., 2018 [119] | 2 | 3 | 3 | 3 | 3 | 3 | 3 | 3 | 2 | 3 | NA | 3 | 3 | NA | 0 | 3 | 37 | 88.1 |
| Kuria et al., 2012 [120] | 1 | 3 | 3 | 2 | 2 | 3 | 3 | 3 | 1 | 3 | NA | 3 | 2 | NA | 0 | 2 | 31 | 73.8 |

(*Continued*)

**Table 1.** (Continued)

| | | | | | | | | | | | | | | | | | | |
|---|---|---|---|---|---|---|---|---|---|---|---|---|---|---|---|---|---|---|
| Kurth et al., 2015 [121] | 1 | 3 | 3 | 2 | 3 | 3 | 3 | 3 | 1 | 3 | NA | 3 | 3 | NA | 0 | 2 | 33 | 78.6 |
| Kurui & Ogoncho, 2019 [122] | 1 | 3 | 3 | 3 | 2 | 2 | 1 | 2 | 0 | 3 | NA | 3 | 1 | NA | 0 | 0 | 24 | 57.1 |
| Kurui & Ogoncho, 2020 [123] | 1 | 3 | 2 | 2 | 3 | 1 | 1 | 1 | 1 | 3 | NA | 3 | 1 | NA | 0 | 0 | 22 | 52.4 |
| Kwamanga et al., 2001 [124] | 0 | 3 | 3 | 3 | 3 | 2 | 1 | 2 | 0 | 3 | NA | 3 | 1 | NA | 3 | 0 | 27 | 64.3 |
| Kwamanga et al., 2003 [125] | 1 | 3 | 3 | 2 | 3 | 2 | 1 | 2 | 0 | 2 | NA | 3 | 1 | NA | 0 | 0 | 23 | 54.8 |
| Kwobah et al., 2017 [126] | 0 | 3 | 3 | 3 | 3 | 3 | 3 | 3 | 2 | 3 | NA | 3 | 2 | NA | 0 | 2 | 33 | 78.6 |
| L'Engle et al., 2014 [127] | 1 | 3 | 3 | 3 | 3 | 3 | 3 | 3 | 2 | 3 | NA | 3 | 3 | NA | 0 | 0 | 33 | 78.6 |
| Lo et al., 2013 [128] | 1 | 3 | 3 | 3 | 3 | 3 | 1 | 3 | 1 | 3 | NA | 3 | 2 | NA | 0 | 3 | 32 | 76.2 |
| Luchters et al., 2011 [129] | 1 | 3 | 3 | 2 | 3 | 3 | 3 | 2 | 2 | 3 | NA | 3 | 3 | NA | 0 | 2 | 33 | 78.6 |
| Lukandu et al., 2015 [130] | 1 | 3 | 1 | 1 | 3 | 3 | 1 | 3 | 0 | 3 | NA | 3 | 1 | NA | 0 | 0 | 23 | 54.8 |
| Macigo et al., 2006 [32] | 1 | 3 | 1 | 1 | 2 | 2 | 0 | 2 | 0 | 3 | NA | 3 | 1 | NA | 0 | 1 | 20 | 47.6 |
| Magati et al., 2018 [131] | 0 | 3 | 2 | 3 | 3 | 1 | 1 | 1 | 0 | 3 | NA | 3 | 3 | NA | 0 | 0 | 23 | 54.8 |
| Maina et al., 2015 [132] | 1 | 3 | 3 | 3 | 2 | 2 | 3 | 2 | 2 | 3 | NA | 3 | 2 | NA | 0 | 2 | 31 | 73.8 |
| Mannik et al., 2018 [133] | 0 | 2 | 3 | 2 | 3 | 2 | 1 | 2 | 2 | 3 | NA | 3 | 2 | NA | 0 | 3 | 28 | 66.7 |
| Maru et al., 2003 [30] | 0 | 3 | 3 | 2 | 3 | 1 | 0 | 1 | 0 | 3 | NA | 3 | 0 | NA | 0 | 0 | 19 | 45.2 |
| Mburu et al., 2018 [134] | 2 | 2 | 2 | 2 | 2 | 1 | 1 | 1 | 1 | 3 | NA | 3 | 3 | NA | 0 | 2 | 25 | 59.5 |
| Medley et al., 2014 [135] | 1 | 2 | 1 | 2 | 3 | 3 | 2 | 2 | 1 | 3 | NA | 3 | 3 | NA | 0 | 2 | 28 | 66.7 |
| Menach et al., 2012 [136] | 1 | 3 | 3 | 2 | 2 | 3 | 2 | 2 | 1 | 3 | NA | 3 | 1 | NA | 0 | 1 | 27 | 64.3 |
| Menya et al., 2019 [137] | 1 | 2 | 3 | 2 | 3 | 3 | 1 | 3 | 0 | 3 | NA | 3 | 3 | NA | 0 | 3 | 30 | 71.4 |
| Micheni et al., 2015 [33] | 0 | 2 | 1 | 1 | 3 | 1 | 0 | 1 | 0 | 3 | NA | 3 | 3 | NA | 0 | 2 | 20 | 47.6 |
| Mkuu et al., 2018 [138] | 1 | 3 | 2 | 3 | 3 | 2 | 2 | 2 | 1 | 3 | NA | 3 | 3 | NA | 0 | 2 | 30 | 71.4 |
| Mohammed et al., 2018 [139] | 0 | 3 | 3 | 3 | 3 | 2 | 2 | 2 | 2 | 3 | NA | 3 | 3 | NA | 0 | 3 | 32 | 76.2 |
| Mokaya et al., 2016 [140] | 1 | 3 | 3 | 3 | 3 | 3 | 3 | 3 | 1 | 2 | NA | 3 | 3 | NA | 0 | 0 | 31 | 73.8 |
| Moscoe et al., 2019 [141] | 1 | 2 | 2 | 3 | 2 | 2 | 1 | 2 | 1 | 3 | NA | 3 | 3 | NA | 0 | 3 | 28 | 66.7 |
| Mundan et al., 2013 [142] | 1 | 2 | 3 | 2 | 2 | 2 | 2 | 2 | 1 | 3 | NA | 3 | 3 | NA | 0 | 3 | 29 | 69.0 |
| Mungai & Midigo, 2019 [143] | 1 | 3 | 3 | 2 | 3 | 1 | 1 | 1 | 0 | 3 | NA | 3 | 0 | NA | 0 | 0 | 21 | 50.0 |
| Muraguri et al., 2015 [144] | 0 | 3 | 2 | 2 | 3 | 2 | 0 | 1 | 0 | 3 | NA | 3 | 0 | NA | 0 | 2 | 21 | 50.0 |
| Muriungi & Ndetei, 2013 [145] | 3 | 3 | 3 | 3 | 3 | 3 | 3 | 3 | 1 | 3 | NA | 3 | 3 | NA | 0 | 2 | 36 | 85.7 |
| Muthumbi et al., 2017 [146] | 1 | 2 | 2 | 2 | 3 | 2 | 0 | 2 | 0 | 3 | NA | 3 | 3 | NA | 0 | 3 | 26 | 61.9 |
| Mutiso et al., 2019 [147] | 3 | 3 | 3 | 3 | 3 | 3 | 3 | 3 | 2 | 3 | NA | 3 | 3 | NA | 0 | 3 | 38 | 90.5 |
| Muture et al., 2011 [148] | 1 | 3 | 2 | 2 | 3 | 2 | 1 | 2 | 0 | 3 | NA | 3 | 3 | NA | 0 | 1 | 26 | 61.9 |
| Mwangi et al., 2019 [149] | 2 | 3 | 3 | 3 | 3 | 3 | 3 | 3 | 1 | 3 | NA | 3 | 3 | NA | 0 | 3 | 36 | 85.7 |
| Nall et al., 2019 [150] | 2 | 3 | 1 | 3 | 3 | 2 | 2 | 2 | 1 | 3 | NA | 3 | 3 | NA | 0 | 3 | 31 | 73.8 |
| Ndegwa & Waiyaki, 2020 [151] | 2 | 3 | 3 | 3 | 2 | 3 | 2 | 2 | 1 | 3 | NA | 3 | 1 | NA | 0 | 0 | 28 | 66.7 |
| Ndetei et al., 2008 [28] | 0 | 3 | 3 | 0 | 3 | 1 | 1 | 1 | 0 | 3 | NA | 3 | 0 | NA | 0 | 0 | 18 | 42.9 |
| Ndetei et al., 2008 [152] | 1 | 1 | 3 | 1 | 2 | 1 | 2 | 1 | 1 | 3 | NA | 3 | 3 | NA | 0 | 0 | 22 | 52.4 |
| Ndetei et al., 2009 [153] | 0 | 3 | 3 | 2 | 3 | 2 | 1 | 2 | 1 | 3 | NA | 3 | 1 | NA | 0 | 0 | 24 | 57.1 |
| Ndetei et al., 2009 [154] | 0 | 3 | 3 | 2 | 3 | 2 | 1 | 2 | 0 | 3 | NA | 3 | 2 | NA | 0 | 0 | 24 | 57.1 |
| Ndetei et al., 2010 [34] | 1 | 2 | 1 | 2 | 2 | 1 | 2 | 1 | 1 | 3 | NA | 3 | 1 | NA | 0 | 0 | 20 | 47.6 |
| Ndetei et al., 2012 [155] | 0 | 3 | 3 | 0 | 0 | 3 | 3 | 3 | 0 | 3 | NA | 3 | 3 | NA | 0 | 3 | 27 | 64.2 |
| Ndugwa et al., 2011 [156] | 3 | 3 | 3 | 3 | 3 | 3 | 3 | 2 | 2 | 3 | NA | 3 | 3 | NA | 0 | 3 | 37 | 88.1 |
| Ng'ang'a et al., 2018 [157] | 1 | 3 | 3 | 3 | 2 | 3 | 3 | 2 | 2 | 3 | NA | 3 | 3 | NA | 0 | 2 | 33 | 78.6 |
| Ngaruyia et al., 2018 [158] | 1 | 3 | 3 | 3 | 3 | 3 | 3 | 3 | 2 | 3 | NA | 3 | 3 | NA | 0 | 2 | 35 | 83.3 |
| Nguchu et al., 2009 [159] | 0 | 3 | 3 | 2 | 2 | 3 | 2 | 3 | 2 | 3 | NA | 3 | 1 | NA | 0 | 1 | 28 | 66.7 |
| Ngure et al., 2019 [160] | 1 | 3 | 2 | 3 | 3 | 2 | 2 | 2 | 2 | 3 | NA | 3 | 1 | NA | 0 | 0 | 27 | 64.3 |
| Nielsen et al., 1989 [161] | 0 | 3 | 1 | 1 | 2 | 2 | 3 | 2 | 3 | 3 | NA | 3 | 1 | NA | 0 | 0 | 24 | 57.1 |
| Njoroge et al., 2017 [162] | 0 | 2 | 2 | 2 | 3 | 2 | 1 | 2 | 0 | 3 | NA | 3 | 3 | NA | 0 | 0 | 23 | 54.8 |
| Njuguna et al., 2013 [25] | 0 | 1 | 3 | 1 | 2 | 1 | 0 | 1 | 0 | 3 | NA | 3 | 0 | NA | 0 | 1 | 16 | 38.1 |

(*Continued*)

**Table 1.** (Continued)

| Study | | | | | | | | | | | | | | | | | | |
|---|---|---|---|---|---|---|---|---|---|---|---|---|---|---|---|---|---|---|
| Ogwell et al., 2003 [163] | 0 | 3 | 3 | 3 | 3 | 3 | 1 | 3 | 0 | 3 | NA | 3 | 3 | NA | 0 | 2 | 30 | 71.4 |
| Okal et al., 2013 [164] | 0 | 1 | 2 | 3 | 3 | 3 | 1 | 3 | 0 | 3 | NA | 3 | 1 | NA | 0 | 3 | 26 | 61.9 |
| Olack et al., 2015 [165] | 1 | 3 | 3 | 2 | 2 | 2 | 3 | 2 | 1 | 3 | NA | 3 | 2 | NA | 0 | 3 | 30 | 71.4 |
| Ominde et al., 2019 [35] | 0 | 3 | 2 | 2 | 2 | 2 | 1 | 1 | 0 | 3 | NA | 3 | 1 | NA | 0 | 0 | 20 | 47.6 |
| Omolo & Dhadphale, 1987 [36] | 0 | 3 | 3 | 2 | 2 | 2 | 0 | 2 | 0 | 3 | NA | 3 | 0 | NA | 0 | 0 | 20 | 47.6 |
| Ongeri et al., 2019 [166] | 1 | 3 | 3 | 3 | 3 | 3 | 3 | 3 | 2 | 3 | NA | 3 | 3 | NA | 0 | 2 | 35 | 83.3 |
| Onsomu et al., 2015[167] | 1 | 2 | 2 | 3 | 3 | 3 | 2 | 2 | 2 | 3 | NA | 3 | 3 | NA | 0 | 2 | 31 | 73.8 |
| Othieno et al., 2000 [168] | 0 | 3 | 2 | 2 | 3 | 2 | 1 | 2 | 1 | 3 | NA | 3 | 1 | NA | 0 | 0 | 23 | 54.8 |
| Othieno et al., 2014[169] | 1 | 3 | 3 | 3 | 2 | 2 | 1 | 2 | 1 | 3 | NA | 3 | 3 | NA | 0 | 2 | 29 | 69.0 |
| Othieno et al., 2015a[170] | 1 | 3 | 3 | 3 | 2 | 2 | 1 | 2 | 1 | 3 | NA | 3 | 3 | NA | 0 | 2 | 29 | 69.0 |
| Othieno et al., 2015b [171] | 1 | 3 | 3 | 2 | 3 | 2 | 2 | 2 | 2 | 3 | NA | 3 | 3 | NA | 0 | 2 | 31 | 73.8 |
| Owuor et al., 2019 [172] | 3 | 3 | 3 | 3 | 3 | 3 | 3 | 2 | 3 | 3 | NA | 3 | 2 | NA | 1 | 1 | 36 | 85.7 |
| Oyaro et al., 2018 [173] | 0 | 3 | 3 | 2 | 3 | 3 | 1 | 2 | 1 | 3 | NA | 3 | 1 | NA | 0 | 0 | 25 | 59.5 |
| Pack et al., 2014 [174] | 1 | 3 | 3 | 2 | 3 | 3 | 2 | 2 | 2 | 3 | NA | 3 | 2 | NA | 0 | 3 | 32 | 76.2 |
| Papas et al., 2011 [175] | 2 | 3 | 3 | 2 | 3 | 3 | 3 | 3 | 2 | 3 | NA | 3 | 3 | NA | 1 | 2 | 36 | 85.7 |
| Papas et al., 2016[176] | 1 | 3 | 3 | 2 | 2 | 3 | 3 | 3 | 2 | 3 | NA | 3 | 3 | NA | 0 | 2 | 33 | 78.6 |
| Papas et al., 2017 [177] | 2 | 3 | 3 | 3 | 2 | 3 | 3 | 3 | 3 | 3 | NA | 3 | 3 | NA | 3 | 3 | 40 | 95.2 |
| Parcesepe et al., 2016 [178] | 1 | 3 | 3 | 2 | 3 | 3 | 3 | 3 | 2 | 3 | NA | 3 | 3 | NA | 0 | 2 | 34 | 81.0 |
| Patel et al., 2013 [179] | 1 | 3 | 2 | 2 | 3 | 1 | 1 | 1 | 0 | 3 | NA | 3 | 1 | NA | 0 | 0 | 21 | 50.0 |
| Peltzer et al., 2009 [180] | 1 | 3 | 2 | 2 | 3 | 3 | 3 | 2 | 2 | 3 | NA | 3 | 3 | NA | 0 | 3 | 33 | 78.6 |
| Peltzer et al., 2011 [181] | 1 | 3 | 1 | 2 | 3 | 3 | 1 | 2 | 1 | 3 | NA | 3 | 3 | NA | 0 | 2 | 28 | 66.7 |
| Pengpid & Peltzer, 2019 [182] | 0 | 3 | 2 | 2 | 3 | 3 | 3 | 2 | 2 | 3 | NA | 3 | 3 | NA | 0 | 2 | 31 | 73.8 |
| Perl et al., 2015 [183] | 1 | 2 | 1 | 2 | 3 | 2 | 1 | 2 | 0 | 3 | NA | 3 | 3 | NA | 0 | 3 | 26 | 61.9 |
| Ploubidis, 2013 [184] | 3 | 3 | 3 | 2 | 3 | 2 | 1 | 2 | 0 | 3 | NA | 3 | 3 | NA | 0 | 3 | 31 | 73.8 |
| Roth et al., 2017 [185] | 1 | 2 | 3 | 2 | 3 | 2 | 1 | 2 | 0 | 3 | NA | 3 | 2 | NA | 0 | 2 | 26 | 61.9 |
| Rudatsikira et al., 2007 [186] | 0 | 2 | 2 | 2 | 3 | 2 | 1 | 2 | 0 | 3 | NA | 3 | 3 | NA | 0 | 2 | 25 | 59.5 |
| Sanders et al., 2007 [187] | 0 | 2 | 2 | 3 | 3 | 2 | 1 | 2 | 0 | 3 | NA | 3 | 3 | NA | 0 | 3 | 27 | 64.3 |
| Saunders et al., 1993 [188] | 1 | 3 | 2 | 2 | 3 | 3 | 2 | 3 | 2 | 3 | NA | 3 | 3 | NA | 0 | 0 | 30 | 71.4 |
| Secor et al., 2015 [189] | 2 | 3 | 2 | 2 | 3 | 3 | 2 | 2 | 2 | 3 | NA | 3 | 2 | NA | 0 | 2 | 31 | 73.8 |
| Syvertsen et al., 2015 [190] | 0 | 3 | 3 | 0 | 0 | 3 | 0 | 2 | 0 | 3 | NA | 3 | 3 | NA | 0 | 3 | 23 | 54.8 |
| Shaffer et al., 2004 [191] | 0 | 2 | 3 | 2 | 3 | 2 | 2 | 2 | 1 | 3 | NA | 3 | 1 | NA | 0 | 1 | 25 | 59.5 |
| Takahashi et al., 2017 [192] | 0 | 3 | 3 | 3 | 3 | 3 | 3 | 3 | 3 | 3 | NA | 3 | 3 | NA | 0 | 3 | 36 | 85.7 |
| Takahashi et al., 2018 [193] | 2 | 3 | 3 | 3 | 2 | 3 | 2 | 3 | 1 | 3 | NA | 3 | 3 | NA | 0 | 2 | 33 | 78.6 |
| Tang et al., 2018 [194] | 1 | 3 | 2 | 1 | 3 | 2 | 1 | 2 | 3 | 0 | NA | 3 | 2 | NA | 0 | 3 | 26 | 61.9 |
| Tegang et al., 2010[195] | 0 | 3 | 3 | 2 | 3 | 3 | 2 | 2 | 3 | 2 | NA | 3 | 3 | NA | 0 | 1 | 30 | 71.4 |
| Thuo et al., 2008 [26] | 0 | 3 | 2 | 1 | 2 | 1 | 1 | 1 | 3 | 0 | NA | 3 | 0 | NA | 0 | 0 | 17 | 40.5 |
| Tsuei et al., 2017 [196] | 0 | 3 | 2 | 2 | 2 | 2 | 1 | 2 | 3 | 0 | NA | 3 | 3 | NA | 0 | 2 | 25 | 59.5 |
| Tun et al., 2015 [197] | 1 | 2 | 3 | 2 | 2 | 3 | 2 | 3 | 3 | 1 | NA | 3 | 2 | NA | 0 | 3 | 30 | 71.4 |
| Wekesah et al., 2018 [198] | 1 | 2 | 3 | 2 | 2 | 2 | 2 | 2 | 3 | 1 | NA | 3 | 3 | NA | 0 | 2 | 28 | 66.7 |
| Were et al., 2014 [199] | 1 | 3 | 2 | 2 | 3 | 2 | 1 | 2 | 3 | 1 | NA | 3 | 1 | NA | 0 | 1 | 25 | 59.5 |
| White et al., 2016 [200] | 1 | 2 | 3 | 2 | 2 | 2 | 2 | 2 | 3 | 1 | NA | 3 | 2 | NA | 0 | 3 | 28 | 66.7 |
| Widmann et al., 2014 [201] | 1 | 3 | 3 | 2 | 3 | 3 | 3 | 3 | 3 | 3 | NA | 3 | 3 | NA | 2 | 3 | 38 | 90.5 |
| Widmann et al., 2017 [202] | 2 | 3 | 3 | 2 | 3 | 3 | 3 | 2 | 3 | 3 | NA | 3 | 3 | NA | 2 | 3 | 38 | 90.5 |
| Wilson et al., 2016 [203] | 1 | 2 | 1 | 3 | 3 | 3 | 3 | 2 | 3 | 3 | NA | 3 | 3 | NA | 3 | 3 | 36 | 85.7 |
| Winston et al., 2015 [204] | 1 | 3 | 2 | 3 | 2 | 3 | 2 | 2 | 3 | 2 | NA | 3 | 3 | NA | 0 | 3 | 32 | 76.2 |
| Winter et al., 2020 [205] | 2 | 3 | 3 | 2 | 2 | 3 | 3 | 2 | 3 | 1 | NA | 3 | 3 | NA | 1 | 3 | 34 | 81.0 |
| Woldu et al., 2019 [206] | 1 | 3 | 3 | 3 | 2 | 3 | 3 | 3 | 2 | 3 | NA | 3 | 3 | NA | 3 | 3 | 38 | 90.5 |

(n = 5), men who have sex with men (MSM) (n = 8), university and college students (n = 9), commercial sex workers (n = 7), psychiatric patients (n = 6), orphans and street connected children and youth (n = 6), people living with HIV (PLHIV) (n = 6), healthcare workers (n = 3), law offenders (n = 3), military (n = 1), and teachers (n = 1). Only one study was conducted among pregnant women [131].

Sixty-nine studies (47.6%) used a standardized diagnostic tool to assess for substance use. The Alcohol Use Disorder Identification Test (AUDIT) (n = 21) and the Alcohol, Smoking & Substance Use Involvement Screening Test (ASSIST) questionnaire (n = 10) were the most frequently used tools. Most papers assessed for alcohol (*n* = 109) and tobacco use (*n* = 80). Other substances assessed included khat (n = 34), opioids (n = 21), sedatives (n = 19), cocaine (n = 19), inhalants (n = 16), cannabis (n = 14), hallucinogens (n = 7), prescription medication (n = 4), emerging drugs (n = 1) and ecstasy (n = 1). Most studies (n = 93) assessed for more than one substance.

**Epidemiology of alcohol use.** One hundred and nine papers assessed for the prevalence and or risk factors for alcohol use. Using the AUDIT, the 12-month prevalence rate for hazardous alcohol use ranged from 2.9% among adults drawn from the community [97] to 64.6% among female sex workers (FSW) [77]. Based on the same tool, the lowest and highest 12-month prevalence rates for harmful alcohol use were both reported among FSWs i.e. 9.3% [80] and 64.0% [174] respectively, while the prevalence of alcohol dependence ranged from 8% among FSWs living with HIV [203] to 33% among MSM who were commercial sex workers [144]. The highest lifetime prevalence rate for alcohol use was reported by Ndegwa & Waiyaki [151]. The authors found that 95.7% of undergraduate students had ever used alcohol.

Alcohol use, was associated with several socio-demographic factors including being male [50, 112, 114, 140, 158, 168, 182, 191], being unemployed [114], being self-employed [97], having a lower socio-economic status (SES) [128], being single or separated, living in larger households [97], having a family member struggling with alcohol use, and alcohol being brewed in the home [143]. Alcohol use was linked to various health factors including glucose intolerance [81], poor cardiovascular risk factor control [111], having a diagnosis of diabetes mellitus [134], hypertension [112, 139], default from tuberculosis (TB) treatment [148], depression [113], psychological Intimate Partner Violence (IPV) [205], tobacco use [182, 205], and increased risk of esophageal cancer [137, 179]. Finally, alcohol use was associated with involvement in Road Traffic Accidents (RTAs) [88], and having injuries [88, 171] and suicidal behavior [109].

**Epidemiology of tobacco use.** Eighty papers assessed for the prevalence and risk factors for tobacco use. The lifetime prevalence of tobacco use ranged from 23.5% among healthcare workers (HCWs) [140] to 84.3% among psychiatric patients [110]. The highest lifetime prevalence rate for tobacco use was reported by Ndegwa & Waiyaki [151]. The authors found that 95.7% of undergraduate students had ever used tobacco.

Tobacco use was associated with socio-demographic factors such as being male [112, 140, 168] and living in urban areas [163]. Several health factors were linked to tobacco use including hypertension [112], development of oral leukoplakia [32], pneumonia [146], increased odds of laryngeal cancer [136], ischemic stroke [100] and diabetes mellitus [134]. In addition, tobacco use was associated with having had an injury in the last 12 months [171], emotional abuse [110], and psychological IPV [205]. Longer duration of smoking was associated with a diagnosis of diabetes mellitus [73], lower SES [128], and hypertension [98, 142]. Peltzer et al. [181] reported that early smoking initiation among boys was associated with ever drunk from alcohol use, ever used substances, and ever had sex. Among girls, the authors found that early smoking initiation was associated with higher education, ever drunk from alcohol use, parental or guardian tobacco use, and suicide ideation.

**Epidemiology of khat use.** The epidemiology of khat use was investigated by 34 studies. The lifetime prevalence rate for khat use ranged from 10.7% among general hospital patients [168] to 88% among a community sample [23]. Khat use was associated with being male [114, 168]; unemployment [114]; being employed [25]; younger age (less than 35 years), higher level of income, comorbid alcohol and tobacco use [166] and age at first paid sex of less than 20 years among FSWs [195]. Further, khat use was associated with increased odds of negative health outcomes [130, 146, 166, 201].

Higher odds of reporting psychotic [166, 201], and PTSD (Post-Traumatic Stress Disorder) symptoms [201], having thicker oral epithelium [130], and pneumonia [146], were reported among khat users compared to non-users.

**Epidemiology of cannabis use.** Fourteen studies evaluated the prevalence of cannabis use. The lifetime prevalence rate of cannabis use ranged from 21.3% among persons with AUD [120] to 64.2% among psychiatric patients [110]. Cannabis use was associated with being male [140, 168], and with childhood exposure to physical abuse [110].

**Epidemiology of opioid and cocaine use.** Twenty-one studies investigated the prevalence of opioid use. The lifetime prevalence rate of opioid use ranged from 1.1% among PLHIV [132] to 8.2% among psychiatric patients [110].

Nineteen studies assessed for the prevalence of cocaine use. The highest reported prevalence rates were 76.2% among PWID use (current use) [190]; 8.8% among healthcare workers (lifetime use) [140]; and 6.7% among PLHIV (lifetime use) [132].

**Epidemiology of IDU.** One study assessed the prevalence for IDU. Key population size estimates for PWID use was reported as 6107 for Nairobi [164]. IDU was associated with depression, risky sexual behavior [149], Hepatitis-C Virus (HCV) infection [173], and HIV-HCV co-infection [68].

**Epidemiology of other substance use (sedatives, inhalants, hallucinogens and prescription medication, emerging drugs, ecstasy).** The epidemiology of sedative use was investigated by 19 studies, inhalant use by 16 studies, hallucinogen use by 7 studies, prescription medication by 4 studies, and emerging drugs and ecstasy by one study each. The highest lifetime prevalence rate for sedative use was reported as 71.4% among a sample of psychiatric patients [28], while the highest prevalence rate for inhalant use was 67% among children living in the streets [86]. The lifetime prevalence rates for hallucinogen use ranged from 1.4% among university students [160] to 3.7% among psychiatric patients [110]. The highest prevalence rate for the use of prescription medication was reported as 21.2% among PWID [190]. One study each reported on the prevalence of emerging drugs [122] and ecstasy [153]. The studies were both conducted among adolescents and youth. The authors found the lifetime prevalence rates for the two substances to be 11.8% [122] and 4.0% [153] respectively.

**Other topics explored by the epidemiology studies.** In addition to prevalence and associated factors, the epidemiological studies explored other topics.

Papas et al. [176] explored the agreement between self-reported alcohol use and the biomarker phosphatidyl ethanol and reported a lack of agreement between self-reported alcohol use and the biomarker phosphatidyl ethanol among PLHIV with AUD.

One study investigated the self-efficacy of primary HCWs for SUD management and reported that self-efficacy for SUD management was lower in those practicing in public facilities and among those perceiving a need for AUD training. Higher self-efficacy was associated with attending to a higher proportion of patients with AUD, and the belief that AUD is manageable in outpatient settings [196].

Five studies investigated the reasons for substance use. Common reasons for substance use included leisure, stress and peer pressure among psychiatric patients[28], curiosity, fun, and peer influence among college students [123], peer influence, idleness, easy access, and curiosity

among adults in the community [25], and peer pressure, to get drunk, to feel better and to feel warm among street children [74]. Atwoli et al. 2011 [72] reported that most students were introduced to substances by friends.

Kaai et al. [99] conducted a study regarding quit intentions for tobacco use and reported that 28% had tried to quit in the past 12 months, 60.9% had never tried to quit, and only 13.8% had ever heard of smoking cessation medication. Intention to quit smoking was associated with being younger, having tried to quit previously, perceiving that quitting smoking was beneficial to health, worrying about future health consequences of smoking, and being low in nicotine dependence. A complete description of the prevalence studies has been provided in Table 2.

## Studies evaluating substance use or SUD programs and interventions

**General description of studies evaluating programs and interventions.** A total of eighteen studies evaluated specific interventions or programs for the treatment and prevention of substance use. These were carried out between 2009 and 2020. Eleven studies focused on individual-level interventions, 5 studies evaluated programs, and 2 studies evaluated population-level interventions. The studies used various approaches including randomized control trials (RCT) (n = 7), mixed methods (n = 3), non-concurrent multiple baseline design (n = 1), quasi experimental (n = 1), cross-sectional (n = 2), and qualitative (n = 3). One study employed a combination of qualitative methods and mathematical modeling.

**Individual-level interventions.** *Individual-level interventions for harmful alcohol use.* Nine studies evaluated either feasibility, acceptability, and or efficacy for individual-level interventions for harmful alcohol use [38, 40, 90, 94, 127, 141, 175, 178, 193]. All the interventions were tested among adult populations including persons attending a Voluntary Counseling & Testing (VCT) center (38), PLHIV [40, 175], and adult males and females drawn from the community [94, 141] and FSWs [127, 178].

Two studies evaluated a six session CBT intervention for harmful alcohol use among PLHIV. The intervention was reported as feasible, acceptable [40] and efficacious [175] in reducing alcohol consumption among PLHIV. The intervention was delivered by trained lay providers.

Giusto et al [90] evaluated the preliminary efficacy of an intervention aimed at reducing men's alcohol use and improving family outcomes. The intervention was delivered in 5 sessions by trained lay-providers, and utilized a combination of behavioral activation, motivational interviewing (MI) and gender norm transformative strategies. The intervention showed preliminary efficacy for addressing alcohol use and family related problems.

Five studies evaluated brief interventions that ranged from 1 to 6 sessions and were delivered by primary HCWs, lay providers and specialist mental health professionals [38, 94, 127, 178, 193]. The brief interventions were reported as feasible, acceptable [38], and efficacious in reducing alcohol consumption [94, 127, 178, 193]. The brief interventions additionally resulted in reductions to IPV, participation in sex work [178], and risky sexual behavior [127].

One study evaluated the efficacy of a mobile delivered MI intervention and found that at 1 month, AUDIT-C scores were significantly higher for waiting-list controls compared to those who received the mobile MI [94].

Moscoe at al. [141] found no effect of a prize-linked savings account on alcohol, gambling and transactional sex expenditures among men.

*Individual-level interventions for khat use.* One study utilized a randomized control trial (RCT) approach to evaluate the effect of a three-session brief intervention for khat use on comorbid psychopathology (depression, PTSD, khat induced psychotic symptoms) and

**Table 2. Studies reporting on the epidemiology of substance use or SUDs.**

| Author, Year | Study design | Study population/ study setting | Sample size | Age; gender distribution | Substance(s) assessed | Standardized tool/criteria used | Main findings (prevalence, risk factors, other key findings) |
|---|---|---|---|---|---|---|---|
| Dhadphale et al. 1982 [12] | Cross-sectional | Students (Secondary school) | 2870 | Age range: 14–20 years Male to female ratio 2:1 | Alcohol, tobacco, cannabis | None | Prevalence of tobacco use 3 or more times a week—16.1% Prevalence of alcohol use 3 or more times a week—10.3% Prevalence of cannabis use was 13.5% at a rate of 1 time per month |
| Omolo & Dhadphale 1987 [36] | Cross-sectional | General patients (Hospital) | 100 | Age distribution not reported Males 50% | Khat[a] | None | Lifetime prevalence khat use was 29%. Mild and moderate chewing significantly associated with age < 20 years (p<0.001) |
| Nielsen et al. 1989 [161] | Cross-sectional | Outpatients (Hospital) | 112 | 18–65 years Males 50% | Alcohol | DSM-III | 30 patients met the criteria for both alcohol abuse and alcohol dependence. 8 patients received a diagnosis of alcohol abuse only and 6 patients received a diagnosis of alcohol dependence only. 39% of the sample exceeded the cut off score for one or both DSM diagnoses |
| Kaplan et al. 1990 [106] | Cross-sectional | Adults (Community) | Not indicated | Age range: 20–40 years Gender distribution not reported | Tobacco | None | Highest prevalence lifetime tobacco smoking was by luo community: 63% males and 67% females Reasons for smoking: positive feelings, to work harder |
| Hall et al. 1993 [93] | Cross-sectional | General patients (Hospital) | 105 | Mean age: 35.4 years Males 78.1% | Alcohol | DSM-III-R/ ICD-I0. | Prevalence of weekly alcohol use was 48% |
| Saunders et al. 1993 [188] | Cross-sectional | General patients (Hospital) | Country specific sample size not reported | Country specific demographics not reported | Alcohol | None | Prevalence of alcohol use for Kenya ->40g per day was 43%, and >60g per day was 37% |
| Dhadphale 1997 [27] | Cross-sectional | Psychiatric patients (Hospital) | 220 | Age range: 18–55 years Males 50.9% | Alcohol | MAST and ICD-9 criteria | Lifetime prevalence of alcohol use among patients with psychiatry morbidity was 12.7% (and 3.1% of those attending outpatient care) |
| Othieno et al. 2000 [168] | Cross-sectional | General patients (Hospital) | 150 | Modal age group: 20–39 years Males 50% | Alcohol, tobacco, khat, cannabis, methaqualone | DSM IV Criteria | Lifetime prevalence: alcohol use (56.7%), tobacco use (32%), khat use (10.7%), cannabis use (5.3%), methaqualone use (0.7%) Alcohol use (p = 0.000), tobacco use (p = 0.000), khat use (p = 0.045), cannabis use (p = 0.004) associated with being male |
| Ayaya et al. 2001 [74] | Case-control | Children living in the streets | 191 | Mean age: 14.03 (SD2.4) Gender distribution not reported | Alcohol, tobacco, cannabis, glue, cocaine | None | Prevalence for drug abuse was 545 per 1000 children. Specific substance prevalence: tobacco 37.6%; sniffing glue 31.2%; alcohol 18.3%; cannabis 8.3%; and sniffing cocaine 4.6% Reasons for substance use included peer pressure, to get drunk, to feel better and to feel warm |
| Kwamanga et al. 2001 [124] | Cross-sectional | Teachers (School) | 800 | Median age: 35 years, Males 74.5% | Tobacco smoking | WHO standard self-administered questionnaire | 50% of males and 3% of females reported tobacco smoking. Peer pressure (63%) and advertisements (21%) are major drivers of smoking |
| Christensen et al. 2009 [81] | Cross-sectional | Adults (Community) | 1179 | Mean age: 38.6 years Males 42% | Alcohol, tobacco | None | Tobacco use was 6.6% in females and 16.2% in males; Alcohol use was 5.4% in females and 20.9% in males Daily alcohol use in males associated with glucose intolerance (p<0.01) |

(*Continued*)

**Table 2.** (*Continued*)

| Author, Year | Study design | Study population/ study setting | Sample size | Age; gender distribution | Substance(s) assessed | Standardized tool/criteria used | Main findings (prevalence, risk factors, other key findings) |
|---|---|---|---|---|---|---|---|
| Kwamanga et al. 2003 [125] | Cross-sectional | Students (Secondary school) | 5311 | Mean age: 16.7 years, Males 68.1% | Tobacco smoking | A WHO standard self-administered questionnaire | Prevalence of current smoking was 10.5%. A total of 12.4% of male students and 6.4% of female students were current smokers. Smoking associated with older age (p<0.001), being in a private school (p<0.001). Reduced odds of stopping smoking with increase in number of tobacco smoked (OR 0.22; 95% CI = 0.19, 0.26; p<0.001) |
| Maru et al. 2003 [30] | Cross-sectional | Children and youth (Juvenile court) | 90 | Age range: 8–18 years Males 71.1% | Alcohol, khat, tobacco, volatile hydrocarbons, sedatives, cannabis | None | Overall prevalence of substance use 43.3%. Tobacco 32.2%; volatile hydrocarbons 21.1%; cannabis 8.9%; alcohol 6.7%; khat 5.6%; sedatives 3.3% Substance use associated with being male (p = 0.0134) |
| Ogwell et al. 2003 [163] | Cross-sectional | Pupils (Primary school) | 1130 | Mean age: 14.1 (SD 0.9) years Males 52% | Tobacco | None | Lifetime tobacco use was 31%, lifetime use of smokeless tobacco was 9%, 55% had friends who smoked Rates of lifetime smoking higher in urban than in suburban students (p<0.005) |
| Astrom et al. 2004 [71] | Cross-sectional | Pupils (Primary school) | 1130 | Mean age: 14.1 (SD 0.9) years Males 52% | Tobacco | None | Tobacco smoking; 31% reported ever smoking tobacco Sources of anti-tobacco messages: broadcast media (47%), Newspapers and magazines (45%), schoolteachers (32%), health workers (29%) |
| Kamotho et al. 2004 [105] | Cross-sectional | Patients undergoing coronary angiography (Hospital) | 144 | Coronary artery disease (CAD): Mean age: 54.4 years, male to female ratio -5.5:1; No CAD: Mean age: 49.8 years, male to female ratio 2.3:1 | Alcohol, tobacco smoking, | None | CAD: Smoking prevalence 15.4%, alcohol 32.7; No CAD: smoking prevalence 13.0%, alcohol 36.9% There was no difference in prevalence of smoking (p = 0.227) and alcohol use (p = 0.67) between those with CAD and those without |
| Shaffers et al. 2004 [191] | Cross-sectional | General patients (Hospital) | 299 | Mean age: 38 (SD 8) years Males 55% | alcohol | AUDIT | Prevalence of hazardous drinking 53.5%, (males 76. %, female 25%), Being male associated with hazardous drinking (p = 0.01) |
| Aden et al. 2006 [23] | Cross-sectional | Adults (Community) | 50 | Age range: 15–34 years Males 80% | Khat | None | Prevalence of khat use was 88% |
| Beckerleg et al. 2006 [76] | Cross-sectional | Adults (Community) | 496 | Age data not given Males 95% | Heroin | None | Prevalence of lifetime heroin injection was 15%; current injection was 7% Average number of years of heroin use was 11.1 years |
| Macigo et al. 2006 [32] | Case-control | Adults and adolescents (Community) | 226 | Age: 15 years and above Males 100% | Tobacco | None | Smoking tobacco was associated with development of oral leukoplakia among those who brushed (RR 4.6 95%CI 2.9–5.1 p<0.001) and those who did not brush teeth (RR 7.3 95%CI 3.6–16.3 p<0.001) |
| Cleland et al. 2007 [82] | Cross-sectional | PWID use (Community) | 106 | Mean age (SD): Males 29 [7]; Female 28 [8] Males 87% | Injection drugs (not specified) | None | Receptive sharing 26% Distributive sharing 41% |

(*Continued*)

**Table 2.** (Continued)

| Author, Year | Study design | Study population/ study setting | Sample size | Age; gender distribution | Substance(s) assessed | Standardized tool/criteria used | Main findings (prevalence, risk factors, other key findings) |
|---|---|---|---|---|---|---|---|
| Kanyanya et al. 2007 [31] | Cross-sectional | Inmates (Prison) | 76 | Mean age: 33.5 years Males 100% | alcohol | DSM-IV Criteria | 71.1% had lifetime abuse or dependence of alcohol |
| Rudatsikira et al. 2007 [186] | Cross-sectional study | Pupils (Primary school) | 242 | Age: 54.3% aged >15 years Gender distribution: males 55.7% | Alcohol, tobacco smoking, other drugs (not specified) | None | Lifetime use: alcohol 10.7%, smoking 10.3%, other drugs 8.4% Past month use: alcohol 9.1%, smoking 6.0% The risk factors for having sex among males were: ever smoked (OR = 2.05, 95%CI 1.92, 2.19), currently drinking alcohol (OR = 1.13, 95%CI 1.06, 1.20), ever used drugs (OR = 2.36, 95%CI 2.24, 2.49) and among females ever used drugs (OR = 2.85, 95%CI 2.57, 3.15). |
| Sanders et al. 2007 [187] | Cross-sectional study | Men who have Sex with Men Exclusively (MSME) and Men who have Sex with both Men and Women (MSMW) (Community) | 285 | Median age (IQR): MSME 27 [23–29]; MSMW 28[23–35] all males | Injection drugs (not specified) | None | Prevalence of IV drug use among MSME was 0.9% and among MSMW was 1.8% |
| Ndetei et al. 2008 [28] | Cross-sectional | Psychiatric Patients (Hospital) | 691 | 78% aged between 21–45 years Males: 63% | Alcohol, opioid, sedatives, khat | SCID-I for DSM IV | Prevalence substance abuse disorder—34.4%. Alcohol use disorder (52%), opiate use disorder (55.5%), sedative use disorder (71.4%), khat use disorder (58.8%) Leisure, stress and peer pressure were the most common reasons given for abusing substances |
| Ndetei et al. 2008 [152] | Cross-sectional | Adults (Community) | 1420 | Mean age: 29.2 years Gender distribution not reported | Alcohol, tobacco, khat, cocaine, heroin, sedatives, opioids, inhalants, phencyclidine, prescription pills, amphetamines | None | Alcohol use prevalence was 36.3% and cocaine 2.2% (most and least abused substances nationally). Prevalence of other substances not stated. Reasons for substance use: leisure, stress and peer pressure |
| Thuo et al. 2008 [26] | Cross-sectional | Psychiatric Patients (Hospital) | 148 | Mean age: 31 years Males nearly two-thirds | Alcohol | SCID for DSM IV | More males (*n* = 39) than females (*n* = 6) were abusing substances (*p*<0.001); Significant associations between PDs and substance abuse dependence (*p*<0.001) |
| Komu et al. 2009 [116] | Cross-sectional | Students (University) | 281 | Age data not given Males 60.4% | Tobacco smoking | None | Prevalence of current tobacco smoking was 12.1% and lifetime prevalence was 38% |
| Ndetei et al. 2009 [153] | Cross-sectional | Students (Secondary school) | 1252 | Mean age: 17 years males 62.5% | Alcohol, tobacco, amphetamines, sedatives, cannabis, hallucinogens, cocaine, methaqualone, ecstasy, heroin, inhalants. | School Toolkit by UNODC | Lifetime smoking reported by 25,3%, daily smoking reported by 3.9% Lifetime use: alcohol 19.6%; heroin 4.0%, amphetamines 18.3%, sedatives 7.0%, cannabis 7.1%, hallucinogen 4.1%, cocaine 4.2%, mandrax 4.0%, ecstasy 4.0%, inhalants 6.6% Age at first use as low as below 11 years |

(*Continued*)

**Table 2.** (Continued)

| Author, Year | Study design | Study population/ study setting | Sample size | Age; gender distribution | Substance(s) assessed | Standardized tool/criteria used | Main findings (prevalence, risk factors, other key findings) |
|---|---|---|---|---|---|---|---|
| Ndetei et al. 2009 [154] | Cross-sectional | Students (Secondary school) | 1328 | Mean age: 16 years Males 58.9% | Not specified | DUSI-R | Prevalence of substance abuse was 33.9% but substances not specified. Substance use associated with psychiatric morbidity, school performance, social competence, peer relations, involvement in recreation, behavior problems (p<0.001 in each case). |
| Nguchu et al. 2009 [159] | Cross-sectional | Patients with diabetes (Hospital) | 400 | Mean age: 63.3 years Males 60% | Tobacco smoking | None | Prevalence of tobacco smoking was 8.4% |
| Peltzer et al. 2009 [180] | Cross-sectional | Students (School) | 2758[c] | 13–15 years Country specific gender distribution not reported | Alcohol, tobacco, illicit drugs (not specified) | Global School-Based Health Survey questionnaire | Prevalence tobacco use 17.5%, illicit drug use 9.5%, risky drinking 4.7% |
| Ndetei et al. 2010 [34] | Cross-sectional | Students (Secondary school) | 343 | Mean age: 16.8 years Males 64.1% | Alcohol, tobacco, cannabis, khat, cocaine, heroin | None | Alcohol, tobacco, khat and cannabis were the most commonly reported substance of use, with user prevalence rates of 5.2%, 3.8%, 3.2%, and 1.7%, respectively. |
| Tegang et al. 2010 [195] | Cross-sectional | FSWs (Community) | 297 | Median age 25 (IQR 21–29) All female | Tobacco smoking, khat, alcohol, heroin | None | Lifetime prevalence:91% for alcohol, 71% for *khat*, 34% for cannabis, and 6% for heroin, cocaine, glue or petrol. Lifetime prevalence of at least one substance was 96%, at least two substances 80% Lifetime use khat associated with age at first paid sex of <20 years (p<0.01); lifetime use tobacco associated with engagement in sex work of >5years (p<0.05); lifetime use heroin/cocaine/ glue/petrol associated with sex with 2 or more partners (p<0.005). |
| Atwoli et al. 2011 [72] | Cross-sectional | Students (University) | 500 | Mean age: 22.9 (SD2.5) Males 52.2% | Alcohol, tobacco | WHO Model Core Questionnaire | Alcohol use: lifetime prevalence was 51.9%; Current prevalence was 50.7%; Among those using alcohol, 50.4% used 5 or more drinks per day, on 1 or 2 days and 9.2% used for3 or more days. Lifetime tobacco smoking was 42.8%; cannabis (2%), cocaine (0.6%). Tobacco use higher among males compared to females (p < 0.05). 75.1% introduced to substances by a friend Reasons for use: to relax (62.2%) or relieve stress (60.8%). |
| Kinoti et al. 2011 [114] | Cross-sectional | Adults (Community) | 217 | Mean age: 34.2 years males 70.5% | Alcohol, khat | None | Prevalence of use for bottled beer: 64.8%; local brew– 41.6%; khat chewing– 41.6%; cannabis -13.7% Males significantly more likely to use bottled beer (p<0.01) and local brew (p<0.01) and khat (p<0.01) Unemployment associated with use of bottled beer (p<0.05) and local brew (p<0.01) and khat (p<0.01) |

(*Continued*)

**Table 2.** (*Continued*)

| Author, Year | Study design | Study population/ study setting | Sample size | Age; gender distribution | Substance(s) assessed | Standardized tool/criteria used | Main findings (prevalence, risk factors, other key findings) |
|---|---|---|---|---|---|---|---|
| Luchters et al. 2011 [129] | Cross-sectional | MSW (Community) | 442 | Mean age: 24.6 (SD 5.2) All males | Alcohol and others (Khat, rohypnol, heroin or cocaine) | AUDIT | Alcohol: overall prevalence of use 70%; 35% of participants who drink had hazardous drinking, 15% harmful drinking and 21% alcohol dependence. Binge drinking prevalence of 38.9% Prevalence of other substances (khat 75.5%, cocaine/heroin 7.7%, rohypnol 14.9%) Alcohol dependence was associated with inconsistent condom use (AOR = 2.5, 95% CI = 1.3–4.6), penile or anal discharge (AOR = 1.9, 95% CI = 1.0–3.8), and two-fold higher odds of sexual violence (AOR = 2.0, 95%CI = 0.9–4.9). |
| Muture et al. 2011 [148] | Case-control | Cases were patients on treatment for tuberculosis (Hospital) | 1978 cases and 945 controls | Mean age/age range: mean 31.2 years for cases and 29.5 years for controls Males 59.4% in cases and 53% of controls | Alcohol | None | Alcohol abuse was found to be a predictive factor for defaulting from TB treatment (OR 4.97; CI 1.56–15.9). |
| Ndugwa et al. 2011 [156] | Cross-sectional | Adolescents living in an informal settlement (community) | 1722 | Mean age: 12–19 years Males 47.2% | Alcohol, tobacco, miraa, glue illicit drugs (not specified) | MPBI | Lifetime prevalence of alcohol use was 6.0%; tobacco smoking was 2.6%; other illicit drugs (not specified) 6.8% |
| Peltzer et al. 2011 [181] | Cross-sectional | Pupils (Primary school) | | Mean age/ range: 13–15 years Gender distribution: 47.7% | Tobacco smoking | GSHS core questionnaire | Lifetime smoking prior to age 14 years reported by 15.5% (20.1% boys and 10.9% girls) early smoking initiation was among boys associated with ever drunk from alcohol use (OR = 4.73, p = 0.001), ever used drugs (OR = 2.36, p = 0.04) and ever had sex (OR = 1.63, p = 0.04). Among girls, it was associated with higher education (OR = 5.77, p = 0.001), ever drunk from alcohol use (OR = 4.76, p = 0.002), parental or guardian tobacco use (OR = 2.83, p = 0.001) and suicide ideation (OR = 2.05, p = 0.02) |
| Embleton et al. 2012 [85] | Cross-sectional | Children living in the streets | 146 | Age range: 10–19 Males 78% | Alcohol, glue, tobacco, cannabis, khat, prescription medication, petrol | None | Lifetime substance use was 74%, current substance use was 62% Lifetime and current prevalence for specific substances respectively was: glue 67%, 58%; alcohol 47%, 16%; tobacco 45% 21%; khat 33%,7%; cannabis 29%,11%; petrol 24%,5%; and pharmaceuticals 8%,<1% Factors associated with having any lifetime drug use were increasing age (adjusted odds ratio [AOR] = 1.47, 95% CI = 1.15–1.87), having a family member who used alcohol, tobacco, or other drugs (AOR = 3.43, 95% CI = 1.15–10.21), staying in a communally rented shelter (AOR = 3.64, 95% CI = 1.13–11.73), and being street-involved for greater than 2 years (AOR = 3.69, 95% CI = 1.22–11.18). |

(*Continued*)

**Table 2.** (Continued)

| Author, Year | Study design | Study population/ study setting | Sample size | Age; gender distribution | Substance(s) assessed | Standardized tool/criteria used | Main findings (prevalence, risk factors, other key findings) |
|---|---|---|---|---|---|---|---|
| Kuria et al. 2012 [120] | Cross-sectional | Persons with alcohol use disorder in an informal settlement (community) | 188 | Mean age: 31.9 years Male 91.5% | Alcohol | CIDI, ASSIST and AUDIT | Tobacco—50% of the participants Cannabis—21.3% There was a statistically significant association (P value 0.002) between depression and the level of alcohol dependence at intake. And at 6 months |
| Menach et al. 2012 [136] | Case-control | Cases were adults with laryngeal cancer (Hospital) | 100 (50 cases, 50 controls) | Mean age: 61years in cases and 63years in control group 96% males | Alcohol, tobacco | None | Being a current smoker increased laryngeal cancer risk with an odds ratio (OR) of 30.4 ($P < 0.0001$; 95% CI: 8.2–112.2). |
| Ndetei et al. 2012 [155] | Cross-sectional | Psychiatric Patients (Hospital) | 691 | Schizoaffective disorder: Mean age 33.1 years; Males 52.2% Schizophrenia mean age: 33.5 years; Males:62.9% Mood disorders: mean age 33.2 years; Males: 58.4% | Alcohol, drugs (not specified) | SCID-I for DSM IV | Comorbidity with alcohol dependence disorder was more common in schizoaffective disorder than with schizophrenia (p = 0.008) |
| Ayah et al. 2013 [73] | Cross-sectional | Adults living in informal settlements (community) | 2061 | Mean age 33.4 years Males 50.9% | Alcohol, tobacco | WHO STEPS survey instrument | Tobacco use Current smoking 13.1% of whom 84.8% were daily smokers. The mean age of smoking commencement and duration of smoking was 19.7 years and 16.5 years Respectively Alcohol use Lifetime prevalence 30%, of whom 74.9% used in past 12 months and 62.2% in the previous 30 days Daily use was 19.7% and use 1–6 days per week among 43.4% Duration of smoking (p = 0.001) and number of pack years(p = 0.049) associated with diagnosis of diabetes |
| Embleton et al. 2013 [86] | Mixed-methods (cross-sectional and qualitative) | Children living in the streets | 146 | Age range: 10–19 years males 85% | Alcohol, glue, tobacco, khat, cannabis, petrol, prescription medication | None | Prevalence of substance use was as follows: glue 67%; alcohol 47%; tobaccos 45%; khat 33%; cannabis 29%; petrol 24%; and pharmaceuticals 8%; |
| khasakala et al. 2013 [108] | Cross-sectional | Youth attending an out-patient clinic (Hospital) | 250 | Mean age: 16.92 years Males 59.1 | Alcohol, other substances (not specified) | MINI (DSM IV) | Any drug use prevalence was 62.4% Alcohol abuse prevalence was 47.8% associations between major depressive disorders and any drug abuse (OR = 3.40, 95% CI 2.01 to 5.76, p < 0.001), or alcohol use (OR = 3.29, 95% CI 1.94 to 5.57, p < 0.001), |

(*Continued*)

**Table 2.** (Continued)

| Author, Year | Study design | Study population/ study setting | Sample size | Age; gender distribution | Substance(s) assessed | Standardized tool/criteria used | Main findings (prevalence, risk factors, other key findings) |
|---|---|---|---|---|---|---|---|
| khasakala et al. 2013 [109] | Cross-sectional | Youth and biological parents attending a youth clinic (Hospital) | 678 (250 youth, 226 biological mothers, 202 biological fathers) | Mean age youth 16.92years males 59.1% (youth) | Alcohol, other substances (not specified) | MINI (DSM IV) | Alcohol use—46.8% of youth, 1.2% mothers and 39.2% of fathers Multiple drug use identified in 9% of youth Significant statistical association between alcohol abuse (p <0.001), substance abuse (p < 0.001) and suicidal behaviour in youths. |
| Kinaynjui & Atwoli 2013 [115] | Cross-sectional | Inmates (Prison) | 395 | Mean age: 33.3 years Males 68.6% | Alcohol, tobacco, cannabis, amphetamines, inhalants, sedatives, tranquillizers, cocaine, heroin. | WHO Model Core questionnaire | Lifetime prevalence of any substance use was 66.1% Lifetime prevalence: alcohol 65.1%, tobacco use 32.7%, tobacco chewing 22.5% admitted to chewing tobacco, cannabis 21%, amphetamines (9.4%), volatile inhalants (9.1%), sedatives (3.8%), tranquillizers (2.3%), cocaine (2.3%), and heroin (1.3%). Substance use associated with male gender (p<0.001), urban residence (p<0.001). |
| Lo et al. 2013 [128] | Cross-sectional | Adults (Community) | 72292 | Modal age group: 18–29 years males 43.1% | Alcohol, tobacco | None | Prevalence of ever smoking was 11.2% and of ever drinking, 20.7%. Percentage of current smokers rose with the number of drinking days in a month (P < 0.0001). Tobacco and alcohol use increased with decreasing socio-economic status and amongst women in the oldest age group (P < 0.0001). |
| Mundan et al. 2013 [142] | Cross-sectional | Military personnel attending a clinic (Hospital) | 340 | Mean age: hypertensives 45.1(SD 7.7); normotensive 40.8 (SD 7.3) Males 91.6% | Alcohol, tobacco | None | Alcohol use in 63% of hypertensive patients and 52.07% of normotensive patients Smoking prevalence was 11% among those with hypertension and 4.2% among normotensives. hypertension associated with daily (P < 0.01) and 1–3 times per week (P < 0.05), consumption of alcohol daily Smoking duration is significantly (P < 0.05) longer among participants with hypertension compared to normotensives. |
| Njoroge et al. 2017 [162] | Cross-sectional study | ART-naïve HIV-1 sero-discordant couples attending a clinic (Hospital) | 196 (99 HIV-infected and 97 HIV-uninfected) | Median age 32 years Males 50% | Tobacco, smoking | None | Smoking: prevalence among those HIV positive was 10% current and past was 22%; among those HIV negative was 11% current and 9% past |
| Njuguna et al. 2013 [25] | Cross-sectional | Adults (Community) | 75 | Mean age: 28.3 Males 100% | Khat | None | Overall prevalence of khat use was 68% Khat use was associated with being employed (OR = 2.8, 95% CI 1.03–7.6) Reasons for starting to chew khat included peer influence (40.4%), idleness (23.1%), easy access to khat (19.2%), and curiosity (17.3%) |

(*Continued*)

**Table 2.** (*Continued*)

| Author, Year | Study design | Study population/ study setting | Sample size | Age; gender distribution | Substance(s) assessed | Standardized tool/criteria used | Main findings (prevalence, risk factors, other key findings) |
|---|---|---|---|---|---|---|---|
| Okal et al. 2013 [164] | A combination of 'multiplier method', the 'Wisdom of the Crowds' (WOTC) method and a published literature review. | MSM, PWID, FSWs (Community) | Not reported | Age and gender distribution data not given | Injection drugs (not specified) | None | Approximately 6107 IDU and (plausibly 5031–10 937) IDU living in Nairobi. |
| Patel et al. 2013 [179] | Case-control | Cases were adults with oesophageal cancer (Hospital) | 159 cases and 159 controls | Mean age for males 56.09 years and females was 54.5 years Males 57.9% | Alcohol, snuff, tobacco smoking | None | Smoking, use of snuff and alcohol were associated with increased risk of esophageal cancer (OR = 2.51, 4.74 and 2.64 respectively) |
| Ploubidis et al. 2013 [184] | cross-sectional | Adults (Community) | 4314 | Mean age: 60.8 years Males 49.2%% | Alcohol, tobacco smoking | None | Prevalence of alcohol was 17.7% and smoking prevalence was 6.8% |
| Balogun et al., 2014 [75] | cross-sectional | Pupils (Primary School) | 3666 | Age range: 13–15 years Males 49.1% | Alcohol | None | Past 30-day alcohol use was 17.9% Lifetime drunkenness was 22.5% Past 30-day alcohol use associated with increased odd sleeplessness; Lifetime drunkenness associated with both depression and sleeplessness |
| Bengston et al., 2014 [77] | Cross-sectional | FSWs (Community) | 818 | Age distribution: 30% aged 18–23 All female | Alcohol | AUDIT | Prevalence of hazardous drinking was 64.6%; harmful drinking was 35.5% Higher levels alcohol consumption associated with having never tested for HIV (PR 1.60; 95% CI: 1.07, 2.40). |
| Chersich et al., 2014 [80] | Cross-sectional | FSWS (Community) | 602 | Mean age: 25.1 years Female 100% | Alcohol | AUDIT | Prevalence of hazardous drinking was 17.3% and harmful drinking was 9.3% Harmful drinking associated with increased odds sexual (95% CI adjusted odds ratio [AOR] = 1.9–8.9) and physical violence (95% CI AOR = 3.9–18.0); while hazardous drinkers had 3.1-fold higher physical violence (95% CI AOR = 1.7–5.6). |
| De Menil et al., 2014 [83] | Cross-sectional | Psychiatric patients (Hospital) | 455 | Mean age/ range: 36.3 years Gender distribution: males 66.4 | Alcohol, other substances (not specified) | None | Prevalence of alcohol use disorder was 21.2% and other drug use was 10.4% |
| Joshi et al., 2014 [98] | cross-sectional | Adults living in informal settlements (Community) | 2061 | Mean age: 33.4 (SD 11.6) years Males 50.9% | Alcohol, tobacco | WHO STEPS | Alcohol use: 30.1% reported lifetime alcohol use; 81% alcohol use in past 12 months; 76.8% reported using alcohol in the past 30 days; harmful use by 52% Tobacco: 13.1% reported current smoking (84% of whom used daily) Current smoking (p = 0.018), years of smoking (p = 0.001) associated with having hypertension |

(*Continued*)

**Table 2.** (*Continued*)

| Author, Year | Study design | Study population/ study setting | Sample size | Age; gender distribution | Substance(s) assessed | Standardized tool/criteria used | Main findings (prevalence, risk factors, other key findings) |
|---|---|---|---|---|---|---|---|
| Medley et al., 2014 [135] | Cross-sectional | PLHIV (Hospital) | 1156 | Mean age: 37.2 Gender distribution not reported | Alcohol | None | Overall, 14.6% of participants reported alcohol use in the past 6 months; 8.8% were categorized as non-harmful drinkers and 5.9% as harmful/likely dependent drinkers. Binge drinking reported in 5.4% |
| Othieno et al., 2014 [169] | Cross-sectional | Students (University) | 923 | Mean age: age 23 (SD4.0) males 56.9% | Alcohol, tobacco | None | Students who used tobacco (p = 0.0001) and engaged in binge drinking (p = 0.0029) were more likely to be depressed |
| Pack et al., 2014 [174] | Cross-sectional | FSW (Community) | 619 | 18 years and older Female 100% | Alcohol | AUDIT Tool not specified for other drug use | Hazardous alcohol use 36.0%; harmful alcohol use 64.0%; other drug use 34.1% |
| Were et al., 2014 [199] | Cross-sectional | PWID (Community) | 61 | Age range: 29–33 years Gender distribution: not reported | Brown sugar, rohypnol, khat, tobacco, cocktail, alcohol, injection drugs (heroin, diazepam) | None | Prevalence of substance use was as follows: 43%, brown sugar 16%, rohypnol 61%, tobacco 61%, khat 26%, cocktail 39%, alcohol 52%; injection drugs heroin 100%, diazepam 18% |
| Widmann et al., 2014 [201] | Case-control | Cases were male khat chewers (Community) | 48 (cases = 33, controls = 15) | Mean age: 34 years for cases, 35.1 for controls Males 100% | Alcohol, khat, tobacco, tranquilizers | MINI | Khat chewers experienced more traumatic event types than non-chewers (*p* = 0.007), more PTSD symptoms than non-chewers (*p* = 0.002) and more psychotic symptoms (p = 0.044). |
| Goldblatt et al., 2015 [91] | Cross-sectional | Children living in the streets | 296 | Age range: 13-21years All males | Alcohol, tobacco, khat, glue, fuel | None | Weekly alcohol use reported by 49%;93% reported weekly tobacco use; and 39% reported weekly Cannabis use; 46% reported lifetime use of glue; 8% reported lifetime inhalation of fuel |
| Hulzelbosch et al., 2015 [96] | Cross-sectional | Persons with hypertension in an informal settlement (Community) | 440 | Age: 35 years and above males 42% | Alcohol, tobacco, khat, glue, fuel | WHO STEPS survey instrument | Tobacco use: current 8.4%, former 11.8% Alcohol use: low 84.8%, moderate 6.8%, high 8.4% |
| Kurth et al., 2015 [121] | Cross-sectional | PWID (Community) | 1785 | Mean age 31.7 years in Coast and 30.4 in Nairobi Males 82.4–89.0% | Injection drugs (heroin) | None | 93% injected heroin in the past 30 days. |
| Lukandu et al., 2015 [130] | Case-control | Cases were dental patients (Hospital) | 42 (34 cases, 8 controls) | mean age 28.9 years all males | Alcohol, khat, tobacco, | None | Oral epithelium thicker in khat chewers compared non-chewers (p<0.05); |
| Maina et al., 2015 [132] | Cross-sectional | PLHIV (Hospital) | 200 | Modal age group 34–41 years (27.4%) males 49.7% | Alcohol, tobacco, cocaine, amphetamines, inhalants, sedatives, opioids, hallucinogens, others (not specified) | ASSIST, ASI | Lifetime prevalence of any substance use was 63.1%; alcohol 94.4%; tobacco 49.7%; cocaine 6.7%; amphetamine type stimulants 19.6%; inhalants 3.4%; sedatives 1.7%; opioids 1.1%; hallucinogens 6.6%; others 4.2% 50.3% wrongly identified the alcohol use vignette problem as stress |
| Micheni et al., 2015 [33] | Cohort | MSM and FSW (Community) | 1425 | Median age was 25 for MSM and 26 for FSW Males 50.9% | Alcohol, injection drugs (not specified) | None | Recent alcohol use was associated with reporting of all forms of assault by MSM [(AOR) 1.8, CI 0.9–3.5] and FSW (AOR 4.4, CI 1.41–14.0), |

(*Continued*)

**Table 2.** (Continued)

| Author, Year | Study design | Study population/ study setting | Sample size | Age; gender distribution | Substance(s) assessed | Standardized tool/criteria used | Main findings (prevalence, risk factors, other key findings) |
|---|---|---|---|---|---|---|---|
| Muraguri et al., 2015 [144] | Cross-sectional | MSM (Community) | 563 | MSM who did not sell sex: 30% in the 35 and older age group; MSM who sell sex: 30.8% in the 25–29 age group Males 100% | Alcohol, illicit drugs (not specified) | AUDIT for alcohol use; tool not specified for illicit substances | 62.9% of MSM who did not sell sex had used illicit drugs in the past 12 months while those who sold sex were 78.7%. Possible alcohol dependence was 21.4% among those who did not sell sex while those who sold sex were 33%. |
| Olack et al., 2015 [165] | Cross-sectional | Adults living in informal settlements (Community) | 1528 | Mean age: 46.7 years Males 42% | Alcohol, tobacco smoking | WHO STEPS survey questionnaire | Prevalence of smoking: Current smokers 8.5% and past Smokers 5.1%; Alcohol: Ever Consumed was 30.4%; In the past 12 months was 17% and In the past 30 days was 6.5% |
| Onsomu et al., 2015 [167] | Cross-sectional | Adult women (Community) | 2227 | Age range not reported Females 100% | Alcohol use in husband | None | 385 of women reported that husband uses alcohol |
| Othieno et al., 2015 [170] | Cross-sectional | Students (University) | 923 | Mean age: age 23 (SD4.0) Males 56.9% | Alcohol, tobacco | None | Alcohol use (p<0.001), binge drinking (p<0.01), tobacco use (p<0.001), were significantly associated with increased odds of having multiple sexual partners. |
| Othieno et al., 2015b [171] | Cross-sectional | Students (University) | 923 | Mean age: age 23 (SD4.0) Males 56.9% | Alcohol, tobacco | None | Prevalence of binge drinking was 38.85%; Tobacco use prevalence not reported Binge drinking and tobacco use were significantly associated with injury in the last 12 months (AOR 5.87 and 4.02, p<0.05, respectively) |
| Secor et al., 2015 [189] | Cross-sectional | MSM Community) | 112 | Median age: 26 years Males 100% | Alcohol, other drugs (not specified) | AUDIT, DAST | Prevalence of hazardous or harmful alcohol use was 45%; prevalence harmful use of other drugs 59.8% Alcohol abuse associated with higher PHQ-9 scores (p = 0.02). |
| Syvertsen et al., 2015 [190] | Cross-sectional | PWID (Community) | 151 | Mean age: 28.8 (SD 6.2) years Males 84% | Alcohol, cannabis, prescription pills, cocaine, heroin | None | Prevalence of substance use was: Alcohol at 92.4%; cannabis at 67.6%; prescription pills at 21.2%; cocaine injection at 76.2%; Heroin injection at 29.1% The mean years of injecting was 6.2; |
| Tun et al., 2015 [197] | Cross-sectional | PWID (Community) | 269 | Median age 31 years Males 92.5% | Injection drugs, cannabis, khat, cocaine, tranquilizers | None | Past month injecting drug use (white heroin- 97%; other 3%); past month use: cannabis -66.5%; Khat- 10.8%; cocaine 3.7%; tranquilizers- 58.0% HIV infection was associated with having first injected drugs 5 or more years ago (aOR, 4.3, p = 0.002), and ever having practiced receptive syringe sharing (aOR, 6.2; p = 0.001) |
| Winston et al., 2015 [204] | Cross-sectional | Children living in the streets | 200 | Mean age: 16 years Males 59% | Alcohol and other drugs (not specified) | None | Prevalence of alcohol use was 45.5%; and any drug use was 77.0% Among females, those with HIV infection more frequently reported drug use (91.7% vs 56.5%, p = 0.02), |

(*Continued*)

**Table 2.** (*Continued*)

| Author, Year | Study design | Study population/ study setting | Sample size | Age; gender distribution | Substance(s) assessed | Standardized tool/criteria used | Main findings (prevalence, risk factors, other key findings) |
|---|---|---|---|---|---|---|---|
| Mokaya et al., 2016 [140] | Cross-sectional | Health care workers (Hospital) | 206 | Mean age: 35.3 years (SD 10.1) Males 36.9% | Alcohol, tobacco, sedatives, cocaine, amphetamine-like stimulants, hallucinogens, inhalants, | ASSIST | Lifetime use was 35.8% for alcohol, 23.5% for tobacco, 9.3% for sedatives, 8.8% for cocaine, 6.4% for amphetamine-like stimulants, 5.4% for hallucinogens, 3.4% for inhalants, and 3.9% for opioids Being male associated with lifetime tobacco (p<0.01), alcohol (p<0.01) and cannabis (p<0.01) use. |
| Papas et al., 2016 [176] | Mixed methods | PLHIV (Hospital) | 127 | Median age 37.0 years (IQR 32.0–43.0) Males 48.2% | Alcohol, kuber, tobacco, cannabis, khat, | AUDIT-C | Prevalence of substance use was as follows: alcohol: ≥6 drinks per occasion at least monthly in the past year was 51.2%; Past 30 days other drug use: Tobacco—25.2%; cannabis—3.9%; khat- 8.7%; kuber -10.2% No agreement between self-reported alcohol use and PETH |
| White et al., 2016 [200] | Cohort | FSW (community) | 405 | Modal age group 40–49 years All female | Alcohol | AUDIT | Hazardous/harmful alcohol use significantly associated with a lower likelihood of self-reported sexual abstinence (aRR 0.58; 95% CI 0.45–0.74) |
| Wilson et al., 2016 [203] | Cross-sectional | FSWs who are PLHIV (hospital) | 357 | Age range: 20–61 years Females 100% | alcohol | AUDIT | Any alcohol use was 48.7%; Among those using 59.1% had drinking behaviour consistent with minimal alcohol use problems, 32.8% moderate problems and 8% had severe alcohol problems or possible alcohol use disorder Women with severe alcohol problems (adjusted odds ratio 4.39, 1.16–16.61) were significantly more likely to report recent intimate partner violence. |
| Embleton et al., 2017 [87] | Cross-sectional | Orphaned and separated children (Community, charitable institutions) | 1365 | Mean age 13.9 years Males 52% | Alcohol, drugs (not specified) | None | Prevalence of alcohol and drug use was 8.9% |
| Goodman et al., 2017 [92] | Cross-sectional | Mothers (Community) | 1976 | Mean age: 38.2 years Females 100% | Alcohol | None | 7.95% reported any alcohol consumption and 5% reported weekly alcohol consumption Physical abuse (OR) = 2; 95% CI: (1–4.2)), emotional neglect (OR = 3.18; 95% CI: (1.47–6.91), and living with someone with a mental illness or depression (OR = 2.14; 95% CI: (1.05–4.34)) during the first 18 years of life significantly increased the odds of reporting weekly alcohol consumption. |
| Jenkins et al., 2017 [97] | Cross-sectional | Adults (Community) | 1147 | Age range: 18–60 years Gender distribution: not reported | Alcohol | AUDIT | Lifetime alcohol use was 14.5% for men and 6.8% for women; Hazardous drinking was 9.5% of men and 2.9% of women. Risk of hazardous drinking was increased in men (OR 0.3, C.I. = 0.17 to 0.58 p < 0.001), people living in larger households (OR 1.8, C.I. = 1.09 to 2.97, p = 0.021), people who were single (OR 1.7, C.I. = 0.92 to 3.04, p = 0.093), and those who are self-employed (OR 1.8, C.I. = 1.04 to 2.99, p = 0.036). |

(*Continued*)

**Table 2.** (*Continued*)

| Author, Year | Study design | Study population/ study setting | Sample size | Age; gender distribution | Substance(s) assessed | Standardized tool/criteria used | Main findings (prevalence, risk factors, other key findings) |
|---|---|---|---|---|---|---|---|
| Kamau et al., 2017 [101] | Cross-sectional | Children and adolescents attending a psychiatry out-patient clinic (Hospital) | 166 | mean age: 13.6 (SD 4.16) years males 56% | Alcohol, tobacco, stimulants, cocaine | KSADS and DSM-IV Criteria | Substance use disorder (30.1%) most prevalent presentation. Prevalence tobacco use -6.0%; Alcohol abuse & dependence—7.2%; cannabis abuse and dependence—14.5%; Stimulant abuse 1.8%; cocaine dependence 0.6% |
| Kimando et al., 2017 [111] | Cross-sectional | Patients with diabetes (Hospital) | 385 | Mean age 63.3 years Males 34.5% | Tobacco smoking, alcohol | None | Tobacco smoking was 23.6%; alcohol prevalence was 26.5% Alcohol influences cardiovascular risk factor control (p<0.001) |
| Kunzweiler et al., 2017 [118] | Cohort | MSM (Community) | 711 | Median age (IQR): 24[21–28] | Alcohol | AUDIT-C | Previously diagnosed HIV-positive and out-of-care status was more likely than HIV-negative status among men who did not report harmful alcohol use (p = 0.28) |
| Kwobah et al., 2017 [126] | Cross-sectional | Adults (Community) | 420 | Median age 34 years, IQR 27–46 Males 48.6% | Alcohol and other substances (not specified) | MINI-7 | Alcohol/ Substance Use Disorders (11.7%). Other substances were not specified. |
| Muthumbi et al., 2017 [146] | Case-control | Cases were patients with pneumonia (Hospital) | 281 cases and 1202 controls | Among the 281 cases: 63% were male and 23% aged 15–24 years. | Alcohol, tobacco, snuff, khat | None | Pneumonia associated current smoking (2.19, 95% CI 1.39–3.70), use of khat (OR 3.44, 95% CI 1.72–7.15), use of snuff (OR 2.67, 95% CI 1.35–5.49) |
| Papas et al., 2017 [177] | Cross-sectional | PLHIV with active alcohol use (Hospital) | 614 | Mean age: Male 40.3, Female 37.5 Male 48.5% | Alcohol | AUDIT-C | Alcohol use not associated with physical and sexual violence among both men (p = 0.434) and women (p = 0.449) |
| Roth et al., 2017 [185] | Cross-sectional | Adult males who use alcohol (community) | 220 | Mean age: 35.2 years all males | Alcohol | None | Drinking alcohol with FSWs associated with ever having commercial sex (p<0.001), fighting with FSWs (p<0.01), being physically hurt by FSWs (p<0.01), physically hurting FSWs (p<0.001), being robbed by FSWs (p<0.001) |
| Takahashi et al., 2017 [192] | Cross-sectional | Adults (Community) | 478 | Mean age: 41 (SD 14) Males: females 41.4% | Alcohol, tobacco | AUDIT | Alcohol: prevalence of current drinking was 31.7% and hazardous drinking was 28.7% Tobacco use prevalence was 14.4% Current (p<0.001) and hazardous alcohol use (p<0.001) associated with being male |
| Tsuei et al.,2017 [196] | Cross-sectional | Health care workers (Hospital) | 206 | Mean age: 35.0 years (SD 10.1) Males 37.2% | Alcohol, tobacco | ASSIST | Prevalence moderate risk alcohol use (3.0%); moderate and high risk tobacco use (11.8% and 0.5%) respectively; moderate risk cannabis use (3.4%) Self-efficacy for SUD was lower in those practicing in public facilities and perceiving a need for AUD training; while higher self-efficacy correlated with a higher proportion of patients with AUD in one's setting, access to mental health worker support, cannabis use at a moderate risk level, and belief that AUD is manageable in outpatient settings. |
| Asiki et al., 2018 [70] | Cross-sectional | Adults living in informal settlements (community) | 1942 | Mean age of women was 48.3 (SD 5.30), and of men was 48.8(SD 5.6) Males 45.6% | alcohol, tobacco, | CAGE | BMI among men negatively associated with current tobacco smoking, |

(*Continued*)

**Table 2.** (Continued)

| Author, Year | Study design | Study population/ study setting | Sample size | Age; gender distribution | Substance(s) assessed | Standardized tool/criteria used | Main findings (prevalence, risk factors, other key findings) |
|---|---|---|---|---|---|---|---|
| Budambula et al., 2018 [78] | Cross-sectional | PWID use, non-injecting drug users, non-drug users, with and without HIV (Community) | 451 | Among PWID (HIV positive): Median age 30.6; Males 45.2% Among PWID (HIV negative): Median age 26.8; Males 64.1% | Injection drugs, non-injection drugs (not specified) | None | Occurrence of early age sexual debut, >1 sexual partners, unprotected sex and history of STIs (all p<0.0001) was significantly higher in HIV-infected PWID use than in non-injection drug users and non-drug users Frequency of bisexuality, homosexuality, sex for police protection, sex for drugs was (all p<0.0001) significantly higher in HIV-infected PWIDs as compared to non-injection drug users and non-drug users |
| Cagle et al.,2018 [79] | Cohort | PLHIVV (Hospital) | 854 | Age: 15 years and above 61% females | Alcohol | AUDIT | CD4 count increase was associated with alcohol use (p = 0.051) following ART initiation in ART naïve patients |
| Gathecha et al., 2018 [88] | Cross-sectional | Adults (community) | 4484 | Age range 18-69years Males 60.3% | Alcohol, tobacco | WHO STEPS survey questionnaire | Smokers (p = 0.001) were significantly more likely to be injured in a road traffic crash. Heavy episodic drinking (p = 0.001) and smoking (p < 0.05) were associated with increased likelihood of occurrence of a violent injury. |
| Kaduka et al., 2018 [100] | Cohort | Patients with stroke (Hospital) | 691 | Median age 60 years Males 42.4% | Tobacco, cocaine | WHO STEPS survey | Tobacco smoking risk factor for ischemic stroke (p < 0.001). |
| Kendagor et al., 2018 [107] | Cross-sectional | Adults (Community) | 4203 | Age range: 18–69 years Males 60% | Alcohol, tobacco | WHO STEPS survey questionnaire | 12.7% reported heavy episodic drinking, Respondents who were separated had three times higher odds of HED compared to married counterparts (OR 2.7, 95% CI 1.3–5.7). Tobacco consumption was associated with higher odds of HED (unadjusted OR 6.9, 95% CI 4.4–10.8) |
| Kiburi et al., 2018 [110] | Cross-sectional | Psychiatric in-patients (Hospital) | 134 | Modal age group 31–40 Males 88.1% | Alcohol, tobacco, opioids, cocaine, amphetamines, inhalants, sedatives, khat | ASSIST | Lifetime: prevalence tobacco 84.3%, alcohol 91.8%, cannabis 64.2%, cocaine 5.2%, amphetamine 3%, inhalants 5.2%, sedatives 22.4%, hallucinogens 3.7%, opioids 8.2%, khat 55.2%; 90% had poly-substance use Emotional abuse significantly predicted tobacco (A.O.R = 5.3 (1.2–23.9) and sedative (A.O.R = 4.1 (1.2–14.2) use. Childhood exposure to physical abuse was associated with cannabis use [A.O.R = 2.9 (1.0–7.9)]. |
| Kimbui et al., 2018 [113] | Cross-sectional | Pregnant adolescents (Hospital) | 212 | Mean age: 17.3 years Males 88.1% | alcohol | AUDIT | 43.9% had used alcohol Depression was associated with ever use of alcohol (p = 0.038), and alcohol dependence (p = 0.004) |
| Korhonen et al., 2018 [117] | Cross-sectional | Gay, bisexual and other MSM (Community) | 1476 | Median age (IQR 22–29), Males 100% | Alcohol, other substances (not specified) | AUDIT, DAST | Prevalence for hazardous alcohol use was 44% and for problematic substance use was 51% Transactional sex was associated with hazardous alcohol use [adjusted prevalence ratio (aPR) 1.34, 95% confidence interval (CI) 1.12–1.60]. Childhood abuse and recent trauma were associated with hazardous alcohol use (aPR 1.36, 95% CI 1.10–1.68 and aPR 1.60, 95% CI 1.33–1.93, respectively), and problematic substance use (aPR 1.32, 95% CI 1.09–1.60 and aPR 1.35, 95% CI 1.14–1.59, respectively). |

(*Continued*)

**Table 2.** (*Continued*)

| Author, Year | Study design | Study population/ study setting | Sample size | Age; gender distribution | Substance(s) assessed | Standardized tool/criteria used | Main findings (prevalence, risk factors, other key findings) |
|---|---|---|---|---|---|---|---|
| Kunzweiler et al., 2018 [119] | Cross-sectional | MSM (Community) | 711 | Median age 24 years Males 100% | Alcohol, other substances (not specified) | AUDIT, DAST | Prevalence of harmful alcohol use was 50.1% and prevalence of moderate substance abuse was 23.8% Depressive symptoms were associated with harmful alcohol use (p<0.01) and moderate substance abuse (p = 0.02) |
| Magati et al., 2018 [131] | Cross-sectional | Adults & adolescents (community) | 43898 | Age range: 15–54 rears females 70.8% | Tobacco | None | Overall smoking and smokeless tobacco prevalence rate was 17.3% and 3.10% respectively among men. Lower rates in women with smoking and smokeless tobacco prevalence at 0.18% and 0.93% |
| Mannik et al., 2018 [133] | Cross-sectional | Adults (Community) | 2865 | Median age 50 years Males 45% | Tobacco | None | The point prevalence of tobacco use was 22%. |
| Mburu et al., 2018 [134] | Cohort | Patients with tuberculosis (Hospital) | 347 | Median age 31years Males 71.8% | Alcohol, tobacco | None | Alcohol use and smoking were associated with DM among TB patients (p<0.200) Number of cigarettes smoked per day and significant risk factors of developing DM among TB patients (p = 0.045) |
| Mkuu et al., 2018 [138] | Cross-sectional | Adults (Community) | 718 | Mean age 36.6 years Males 86% | Alcohol, tobacco smoking | AUDIT | An average of 2.5 drinking events and 4.3 binge-drinking occasions per month. 37% consumed unrecorded alcohol. Those who completed primary education or above less likely to report consuming unrecorded alcohol compared to those with incomplete primary education or lower, (OR = 0.22, 95% CI: 0.12–0.43). Compared to poorest and poor respondents, those identifying as middle class or above were less likely to consume unrecorded alcohol (OR = 0.47, 95% CI: 0.29–.78). Current smokers (OR = 2.19, 95% CI: 1.34–3.60) and those with higher binge drinking occasions in the past month (OR = 1.03, 95% CI: 1.004–1.07) were significantly more likely to consume unrecorded alcohol. |
| Mohammed et al., 2018 [139] | Cross-sectional | Adults (Community) | 4484 | Modal age group 18–29 (46%); Gender distribution: not reported | Alcohol, tobacco | WHO STEPS survey questionnaire | Prevalence of current tobacco use was 13.4% and harmful alcohol use was 14.4%. Harmful alcohol use was associated with hypertension (p < 0.001). |
| Ng'ang'a et al., 2018 [157] | Case-control | Cases were women screened for cervical cancer (Community) | 1180 (194 cases, 986 controls) | Age range: 30–49 years Females 100% | alcohol, tobacco | None | Those with binge drinking more likely to be screened for cervical cancer [OR 5.94, 95%CI 1.52–23.15) p = 0.010] |
| Ngaruiya et al., 2018 [158] | Cross-sectional | Adults (Community) | 4484 | Age range: 18–69 years males 48.7% | Alcohol, tobacco | WHO STEPS survey questionnaire | Prevalence of tobacco use: current use was 13.5%; Lifetime alcohol use was 43.1% Men had nearly seven times higher odds of being tobacco users as compared to women (OR 7.63, 95% CI 5.63–10.33). current tobacco use associated with ever use alcohol (p<0.001) |

(*Continued*)

**Table 2.** (Continued)

| Author, Year | Study design | Study population/ study setting | Sample size | Age; gender distribution | Substance(s) assessed | Standardized tool/criteria used | Main findings (prevalence, risk factors, other key findings) |
|---|---|---|---|---|---|---|---|
| Oyaro et al., 2018 [173] | Cross-sectional | PWID (Community) | 673 | Majority between 20–34 years Males 93% | Injection drugs (not specified) | None | IDU was positively associated with HCV (aOR = 5.37, 95% CI:2.61–11.06; p < 0.001) |
| Tang et al., 2018 [194] | Cross-sectional | Adult men (Community) | 12815 | Mean age: 30 (SD 10.9) Males 100% | Tobacco | None | Trends in tobacco use: the rates declined from 22.9% in 2003 to 18.8% in 2008–2009 and 17% in 2014. |
| Wekesah et al., 2018 [198] | Cross-sectional | Adults (community) | 4066 | Age: 18 years and above Male: 48.6% | Alcohol, tobacco | WHO STEPS survey questionnaire | Prevalence of smoking was 10.2% (17.9% males, 2.9% of females) Prevalence of harmful alcohol use 13.8% (24.5% of males and 3.7% of females) |
| Akiyama et al., 2019 [68] | Cross-sectional | PWID and illicit drug use (NSP sites within the community) | 2188 | Median age (IQR): 32 years (28–36) Males 91% | Injection drugs, illicit drugs (not specified) | None | Median (1QR) age at first injection 27 years (24–31), Median (1QR) number of injections per day in the past month: 2 (1–3); Median (1QR) years injecting 3(2–6) Needle sharing at last injection: receptive (3%); distributive (3%) More years of injecting and more injections in the past month was associated with increased odds of HIV–HCV co-infection (p>0.0001 in both cases) |
| Anundo 2019 [69] | Cross-sectional | Female PWID (Community) | 149 | Age range: 26–40 years Females 100% | Alcohol, tobacco, khat, heroin amphetamines, cocaine, hallucinogens, sedatives. | ASSIST | The substance specific risk scores for frequently used substances were as follows: heroin 38, tobacco 37, alcohol 35, khat 28, rohypnol 1, cocaine 1 |
| Gitatui et al. 2019 [24] | Cross-sectional | Adults living in informal settlements (Community) | 215 | Age: above 18 years Males 80% | Alcohol | None | Alcohol use reported on average 4.15 ± 2.8 (Mean ± SD) days per week. Respondents who consumed more than three drinks were more likely (p < 0.05) to be older (OR = 5.8, 95% CI:2.3–14.2 and OR = 2.6, 95% CI: 1.1–6.4), married (OR = 8.3, 95% CI: 3.3–21.1), separated/divorced/widowed (OR = 2.8, 95% CI: 1.3–6.5), had attained post primary education (OR = 2.1, 05% CI: 1.1–3.8), and of income above 50 USD (OR = 5.8, 95% CI: 2.5–13.8 and OR = 8.8, 95% CI: 3.1–25.5) |
| Haregu et al. 2019 [95] | Cross-sectional | Adults living in informal settlements (Community) | 5190 | Age: 18 years and above Males 53.8% | Alcohol, tobacco | None | lifetime alcohol use was 16.4%; lifetime tobacco use 20.3% |
| Kaai et al. 2019 [99] | Cross-sectional | Adult smokers (Community) | 1103 | Age: 18 years and above males 91.5% | Tobacco | None | Quit intentions: 28% had tried to quit in past 12 months; 60.9% had never tried to quit, only 13.8% had ever heard of smoking cessation medication Factors associated with quit intentions: being younger (AOR 3.29 [18–24 years]; AOR 1.98 [25–39 years]), having tried to quit previously (AOR 3.63), perceiving that quitting smoking is beneficial to health (AOR 2.23 [moderately beneficial]; AOR 3.72 [very/extremely beneficial]), worrying about future health consequences of smoking (AOR 3.10 [little/ moderately worried]; AOR 4.05 [very worried]), and being low in nicotine dependence (AOR 0.74). |

(*Continued*)

**Table 2.** (Continued)

| Author, Year | Study design | Study population/ study setting | Sample size | Age; gender distribution | Substance(s) assessed | Standardized tool/criteria used | Main findings (prevalence, risk factors, other key findings) |
|---|---|---|---|---|---|---|---|
| Kamenderi et al. 2019 [102] | Cross-sectional | Students (Secondary schools) | 3908 | Age data not stated males 60% | Alcohol, khat, prescription medication, tobacco, cannabis, inhalants, heroin, cocaine | None | Lifetime use; alcohol (23.4%), khat (17.0%), prescription medication (16.1%), tobacco (14.5%), cannabis (7.5%), inhalants (2.3%), heroin (1.2%) and cocaine (1.1%); |
| Kamenderi et al. 2019 [103] | Cross-sectional | Adolescents and adults (Community) | 3362 households | Age range: 15–65 years Gender distribution not reported | Alcohol, tobacco, cocaine, heroin, khat, | None | Lifetime prevalence of any substance was 62.5%; alcohol use disorder at 10.4%, tobacco use disorder at 6.8%, khat use disorder at 3.1 and heroin use disorder at 0.8% |
| Kamenderi et al. 2019 [104] | Cross-sectional | Adults and adolescents (community) | 2136 households | Mean age/age range: range 15–65 Males 48.8% | Alcohol, tobacco, khat | DSM V Criteria | Prevalence of multi- substance use was 5.3%; Multiple substance use disorder pattern was as follows; alcohol and tobacco (2.5%); tobacco and khat (0.8%), alcohol and khat (0.7%); alcohol, tobacco and khat (0.5%); alcohol, tobacco, khat and bhang (0.3%), alcohol, khat and bhang (0.2%), alcohol, tobacco and bhang (0.2%); alcohol and bhang (0.1%). Predictors of multiple substance use disorder were: setting (more in urban versus rural area) p = 0.004 and gender (more in females) p = 0001 |
| Kimani et al. 2019 [112] | Cross-sectional | Patients with hypertension (Hospital) | 229 | Modal age group: <50 years (40.2%) Males 44.5% | Alcohol, tobacco smoking, | None | Prevalence of tobacco smoking 8.3% and alcohol use 13.1% More males reported drinking alcohol and smoking (p<0.001). Higher BPs were observed in smokers and drinkers (p<0.05). |
| Kisilu et al. 2019 [29] | Cross-sectional | Persons on MMT (MMT clinics) | 388 | Age distribution not reported Males 93% | Alcohol, tobacco, khat, heroin, benzodiazepine, amphetamines, cocaine, barbiturates. | None | Type of substance first used: Cannabis 35.9%, tobacco 29.1%, alcohol 12%, heroin 11.3%, khat 5.9%, benzodiazepine 3%; glue 1.5%, amphetamines 0.3%, cocaine 0.3% and barbiturates 0.2%. |
| Kurui & Ogoncho 2019 [122] | Cross-sectional | Students (College) | 303 | Mean age: 21.96 years Males 49.5% | Alcohol, tobacco, khat, heroin, prescription drugs, emerging drugs (shisha, kuber, shashaman, others not specified) | None | Lifetime use of any substance 52.5%; alcohol 52.5%, Tobacco 12.2%, khat 17.5%, heroin 1.3%, prescription drug 12.5%, emerging drugs 11.2% |
| Menya et al. 2019 [137] | Case-control | Patients with esophageal cancer (Hospital) | 836 (422cases, 414 controls) | Mean age 60 years Males 65% in cases and 61% in control | Alcohol, tobacco | None | For the same amount of ethanol intake, drinkers who had 10 percentage points more ethanol consumed as chang'aa had a 16% (95%CI: 7, 27) higher esophageal squamous cell carcinoma risk. |
| Mungai & Midigo 2019 [143] | Cross-sectional | Adults (Community) | 385 | Age range: 18–65 years Males 62.6% | alcohol | AUDIT | Alcohol use: 65% had hazardous or harmful drinking Harmful/hazardous alcohol use associated with having a family member struggling with alcohol use (p<0.001), alcohol being brewed in the home (p<0.001) |

(*Continued*)

**Table 2.** (Continued)

| Author, Year | Study design | Study population/ study setting | Sample size | Age; gender distribution | Substance(s) assessed | Standardized tool/criteria used | Main findings (prevalence, risk factors, other key findings) |
|---|---|---|---|---|---|---|---|
| Mutiso et al. 2019 [147] | Cross-sectional | Students (Secondary schools) | 471 | Mean age was 16.33 Males 46.5% | Substances not specified | DUSI-R | No significant differences in the mean scores for substance use problems across all the categories, though the lowest scores were reported among those who had not experienced bullying problems |
| Mwangi et al. 2019 [149] | Cross-sectional | PWID, women (Community) | 306 | Mean age 30 years (SD 5.7) Females 100% | Injecting drugs (not specified) | DSM-5 Criteria | 88% of participants had severe injecting drug use (IDU) IDU and depression were related to each other (P < 0.05) and each of them with risky sexual behavior (P < 0.05). |
| Nall et al. 2019 [150]] | Cross-sectional | Youth (Community) | 651 | Mean age: 16.7years Males 46.5% | Alcohol, tobacco | CRAFFT | A mean score of 1.39 (SD 0.81) with 30.4% having a score of two or more on CRAFFT, which is the threshold for intervention Substance use predicted intent to test for HIV, (OR = 1.41, p = 0.007.) |
| Ngure et al. 2019 [160] | Cross-sectional | Students (University) | 1438 | Age range: 17–33 years Males 53% | Opioids, alcohol, tobacco, shisha, kuber[b], khat, inhalants, amphetamines, cocaine, hallucinogens, sedatives | ASSIST | Lifetime prevalence of any substance was 48.6% and current prevalence was 37.9% Lifetime prevalence of tobacco -13%, shisha 17.8%, kuber 4.3%, alcohol 43.2%, 14.2%, cocaine 2.7%, amphetamines 1.7%, inhalants 0.8%, sedatives 0.8%, hallucinogens 1.4%, opioids 1.3%, khat 11.5%, muguka 8.1% |
| Ominde et al. 2019 [35] | Cross-sectional | In-patients with stroke (Hospital) | 227 | Mean age: 68.8 (SD 6.8) Males 37.9% | Alcohol, tobacco | None | Prevalence for alcohol use was 63% and tobacco use was 48% |
| Ongeri et al. 2019 [166] | Cross-sectional | Adults (Community) | 831 | Mean age: 30 years Males 47.6% | Khat, tobacco, alcohol, other drugs (not specified) | ASSIST | Khat: lifetime use 44.6%, current use 36.8% Khat use associated with higher odds of reporting strange experiences (OR, 2.45; 95% CI, 1.13–5.34) and experiencing hallucinations (OR, 2.08; 95% C.I, 1.06–4.08) Khat use significantly associated with male sex (p < 0.001), younger age (less than 35 years) (p < 0001), higher level of income (p < 0.001) and comorbid alcohol (p = 0.001) and tobacco use (p < 0.001). |
| Owuor et al. 2019 [172] | Cross-sectional | Students (University) | 404 | Mean age: 22.42 (SD 2.45) Males 54.8% | Alcohol, tobacco, sedatives, others (not specified) | ASSIST | Lifetime use of at least one substance was 76% and current use was 46.3%. |
| Pengpid & Peltzer 2019 [182] | Cross-sectional | Adults (Community) | 4469 | Median age (38 years) Males 39.7% | Alcohol | WHO STEPS survey questionnaire | 12.8% reported past month binge-drinking and 6.7% had hazardous or harmful alcohol use. Current tobacco and khat use was 12.8% and 6.8% respectively Being male (AOR 7.66 [3.92, 14.97]), tobacco use (AOR 6.72 [3.69, 12.2]), and having hypertension (AOR 2.28 [1.49, 3.48]) increased the odds for hazardous or harmful alcohol use. |
| Woldu et al. 2019 [206] | Cross-sectional | Adults living in informal settlements (community) | 413 | 18 years and older | Alcohol, tobacco, cannabis, khat, cocaine, opioids, sedatives, hallucinogens | ASSIST | Use of any substance in past three months increased the odds of having concurrent sexual relationships (aOR 2.46; 95% CI 1.37–4.42, p < .01). |

(*Continued*)

**Table 2.** (Continued)

| Author, Year | Study design | Study population/ study setting | Sample size | Age; gender distribution | Substance(s) assessed | Standardized tool/criteria used | Main findings (prevalence, risk factors, other key findings) |
|---|---|---|---|---|---|---|---|
| Kamenderi et al. 2020 | Mixed methods (cross-sectional and qualitative) | Pupils (Primary school) | 3307 | Age distribution not reported Males 51.8% | Alcohol | None | Prevalence of alcohol use was 7.2% |
| Kurui & Ogoncho 2020 [123] | Cross-sectional | Students (College) | 303 | Mean age: 21.96 (SD 0.4) years Males 49.5% | Alcohol | None | Prevalence of lifetime alcohol use was 52.5% and current alcohol use was 27.4% Reasons for using alcohol included curiosity 24.1%, fun 12.2%, peer influence 11.6%; Average use- 1 unit 15.2%, 3–4 units 13.2% |
| Mutai et al. 2020 [39] | Mixed methods (cross-sectional and qualitative) | Adults living in informal settlements (community) | 200 | Modal age group 18–24 (74%) Males 60% | Alcohol, khat, kuber, heroin, tobacco | None | Prevalence of substance abuse: Cannabis 60%; alcohol 26.5%; khat 6%; kuber, heroin and tobacco 3% each |
| Ndegwa & Waiyaki 2020 [151] | Cross-sectional | Students (University) | 407 | Age range: 18–41 Males 41.3% | alcohol, tobacco | ASSIST | Tobacco use was reported by 95.7% (77.9% had low risk, 16.3% moderate risk and 1.5% high risk); Alcohol was reported by 95.7% (77.2% low risk; (16.0%) moderate risk; (2.5%) high risk: |
| Winter et al. 2020 [205] | Cross-sectional | Adults living in an informal settlement (community) | 361 | Modal age group: 25–44 years (80%) Female 100% | Alcohol, tobacco | None | Alcohol prevalence was 21.1%, Tobacco prevalence 7.8% Recent psychological IPV was associated with alcohol (OR = 2.6, p<0.05) and tobacco use (OR = 3.8, p<0.05) |

[a]khat *(catha edulis)* is a plant with stimulant properties and is listed by WHO as a psychoactive substance. Its use is common in East Africa

[b]kuber is a type of smokeless tobacco product.

everyday functioning. The intervention was delivered by trained college graduates and was found to result in reduced khat use and increased functioning levels, but had no benefit for comorbidity symptoms (compared to assessments only) [202].

*Individual level intervention for any substance use.* One study evaluated the efficacy of a four-session psychoeducation intervention using an RCT approach. The study found that the intervention was effective in reducing the severity of symptoms of any substance abuse at 6 months compared to no intervention. The intervention was additionally effective in reducing symptoms for depression, hopelessness, suicidality, and anxiety [145].

**Programs.** *Methadone programs.* Two studies utilized qualitative methods to evaluate the perceptions of persons receiving methadone on the benefits of the programs [61, 62]. The methadone programs were perceived as having potential to aid in recovery from opioid use and to reduce HIV transmission among PWID [61, 62].

*Needle-syringe programs (NSPs).* One paper explored the impact of NSPs programs on needle and syringe sharing among PWID. The study reported that the introduction of NSPs led to significant reductions in needle and syringe sharing [56].

*Tobacco cessation programs.* One study evaluated HCWs knowledge and practices on tobacco cessation and found that the knowledge and practice on tobacco cessation was inadequate [89].

*Out-patient SUD treatment programs.* One paper investigated the impact of community based outpatient SUD treatment services and reported a 42% substance use abstinence rate 0–36 months following treatment termination [84].

**Population-level interventions.** *Population-level tobacco interventions.* One study evaluated the appropriateness and effectiveness of HIC anti-tobacco adverts in the African context and found the adverts to be effective and appropriate [183].

*Population-level alcohol interventions.* One paper examined community members' perspectives on the impact of the government's public education messages on alcohol abuse and reported that the messages were ineffective and unpersuasive [55].

A complete description of studies investigating programs and interventions is in Table 3.

## Studies qualitatively exploring various substance use or SUD topics (other than interventions)

**General description of qualitative studies.** There were 23 qualitative studies included in our review. The studies were conducted between 2004 and 2020. Data was collected using several approaches including in-depth interviews (IDIs) only (n = 6), focus group discussions (FGDs) only (n = 2), a combination of FGDs and IDIs (n = 10), a combination of observation and individual IDIs (n = 2), a combination of observation, IDIs and FGDs (n = 1), a combination of literature review, observation, IDIs and FGDs (n = 1). One study utilized the participatory research and action approach [60]. The target populations for the qualitative studies included persons using heroin (n = 3), males and females with IDU (n = 11) adolescents and youth (n = 3), FSWs (n = 2), refugees and Internally Displaced Persons (IDPs) (n = 1), and PLHIV (n = 2).

**Injecting drug use and heroin use.** Thirteen studies explored various themes related to IDU and heroin use with most of them (n = 8) focusing on issues related to women. Three studies explored the drivers of IDU among women and found them to include influence of intimate partners [48, 49], stress of unexpected pregnancies [49], gender inequality, and social suffering [67]. One study found that IDU among women interfered with utilization of antenatal and maternal and child health services [57], while another reported that women who inject drugs linked IDU to amenorrhea hence did not perceive the need for contraception [51].

Mburu et al [47] explored the social contexts of women who inject drugs and found that these women experienced internal and external stigma of being injecting drug users, and external gender-related stigma of being female injecting drug users. Using a socio-ecological approach, Mburu et al [50] reported that IDU during sex work was an important HIV risk behavior. In another study, FSWs reported that they used heroin to boost courage to engage in sex work [65].

Other than IDU and heroin use among women, five studies investigated other themes. One study explored the experiences of injecting heroin users and found that the participants perceived heroin injection as cool [42]. Guise et al. 2015 [44] conducted a study to explore transitions from smoking to injecting and reported that transitions from smoking to IDU were experienced as a process of managing resource constraints, or of curiosity, or search for pleasure. One study explored the experiences of persons on MMT as regards integration of MMT with HIV treatment. The study was guided by the material perspective in sociology theory and Annmarie's Mol's analysis of logic of care. Persons on MMT preferred that they have choice over whether to seek care for HIV and MMT in a single, or in separate settings.

**Alcohol use.** Six studies focused on alcohol use. Three studies explored perceptions of service providers and communities on the effects of alcohol use. Alcohol use was perceived as having a negative impact on sexual and reproductive health [53, 54] and on socio-economic status [43, 46]. One study explored the reasons for alcohol use among PLHIV and found that reasons for alcohol use included stigma and psychological problems, perceived medicinal value, and poverty [60].

**Table 3. Studies evaluating substance use or SUD interventions and programs.**

| Author, Year | Study design | Study objective | Sample size | Name of intervention/ program | Intervention delivered by | Outcomes and measures | Main Findings |
|---|---|---|---|---|---|---|---|
| **Individual-level interventions for harmful alcohol use:** | | | | | | | |
| Mackenzie et al. 2009 [38] | Mixed methods | Evaluate feasibility of an alcohol screening and brief intervention for adult clients attending HIV VCT centres | Intervention group: 456 Comparison group: 602 | 5–10 minute brief intervention. | Trained VCT service providers | • Acceptability<br>• Change in AUDIT scores<br>• Proportion of respondents screened for alcohol use and offered feedback | Intervention feasible and acceptable |
| Papas et al. 2010 [40] | Mixed methods | Cultural adaptation and pilot testing of CBT for alcohol use among HIV-infected outpatients | Focus group 1; 8 Focus group 2; 27 | 6 sessions of CBT delivered by non-professionals | Paraprofessionals | • Treatment attendance<br>• Treatment acceptability, -- Alcohol use assessment using the TLFB method | Culturally adapted CBT was feasible, acceptable, and demonstrated preliminary efficacy |
| Papas et al. 2011 [175] | RCT | Efficacy of CBT for HIV-infected outpatients with hazardous/ binge drinking alcohol | 75 | 6 weekly CBT sessions Control: Usual care | Paraprofessionals | Percent drinking days and mean drinks per drinking days measured using the TLFB method | CBT efficacious |
| Harder et al. 2020 [94] | RCT | To test the effectiveness of a MI intervention using the mobile phone among adults with alcohol use problems. | Intervention group: 89 Control group 1: 65 Control group 2: 76 | Mobile MI–single session MI delivered via mobile phone call upon enrolment Control 1: in-person MI Control 2: delayed mobile MI | Three clinicians with Master's degree in nursing, doctoral degree in clinical psychology and a medical degree | Change in AUDIT-C scores | AUDIT-C scores significantly higher for waiting-list controls after 1 month of no intervention versus mobile MI 1 month after intervention. no difference between in-person and mobile MI at 1 month |
| Moscoe et al. 2019 [141] | RCT | To evaluate the effect of prize-linked savings accounts on men's expenditure on alcohol use and risky sexual behaviors | Intervention: group: 152 Control group: 148 | Intervention: Reward for saving any amount in the bank Control: No reward standard interest | - | Whether a participant saved any money in the bank account during the study period; total amount saved in the bank account; expenditures on alcohol, gambling, and transactional sex. | The intervention did not have a significant effect on alcohol, gambling, and transactional sex expenditures. |
| Giusto et al. 2020 [90] | Non-concurrent multiple baseline design | To evaluate the preliminary efficacy of an intervention aimed at reducing men's alcohol use and improving family outcomes | 9 | 5 session brief intervention combining behavioral activation, MI and gender norm transformative strategies Control: None | Trained lay counselors | Changes in daily alcohol use (TLFB) Changes in PHQ-9 scores Changes in family-oriented behavior | Intervention showed preliminary efficacy for addressing alcohol use and family-related problems |

(*Continued*)

**Table 3.** (Continued)

| Author, Year | Study design | Study objective | Sample size | Name of intervention/program | Intervention delivered by | Outcomes and measures | Main Findings |
|---|---|---|---|---|---|---|---|
| L'Engle et al. 2014 [127] | RCT | Efficacy of a brief intervention for harmful alcohol use for female sex workers | Intervention group: 410 Control group: 408 | Intervention group: 6 counselling sessions based on WHO Brief Intervention for alcohol use Control: 6 sessions Nutritional counselling | Trained nurses | Difference in AUDIT scores and laboratory STI results between intervention and control groups | Intervention efficacious in reducing alcohol use and risky sexual behavior. |
| Parcesepe et al. 2016 [178] | RCT | To document the impact of an alcohol harm reduction intervention on IPV and engagement in sex work among FSWs | Intervention group: 410 Control group: 408 | Intervention: 6 sessions of contextualized WHO Brief Intervention Control: 6 sessions of non-alcohol related nutrition intervention | Trained nurses | Differences in interpersonal violence and engagement in sex work between intervention and control groups | Intervention resulted in reduction in IPV, reduction in sexual partners and reduction in participation in sex work |
| Takahashi et al. 2018 [193] | 3-arm quasi experimental | To assess the effectiveness of community-based alcohol brief intervention with and without motivational talks by former drinkers, in reducing harmful and hazardous alcohol use | Control group: 52 Intervention group 1: 52 Intervention group 2: 57 | Intervention 1: 3 sessions brief intervention based on FRAMES model Intervention 2: 3 sessions BI plus group Motivational talks Control: general health information on alcohol consumption. | Trained community-health workers | Differences in the mean AUDIT scores between the control group and each of the intervention groups at 1, 3 and 6 months, | Greater reduction in adjusted mean AUDIT scores in intervention groups compared to controls |
| **Individual-level interventions for khat use** | | | | | | | |
| Widmann et al. 2017 [202] | RCT | To evaluate impact of a brief intervention for khat use on comorbid psychopathology (depression, PTSD, khat induced psychotic symptoms) and everyday functioning | Intervention group: 161 Control group: 169 | Intervention: 3 sessions Screening and Brief Intervention Control: Assessments for comorbidity and SBI after 2 months | Trained college graduates | Differences in PHQ-9; Post-traumatic diagnostic Scale, ASSIST and everyday functioning scores | Intervention reduced khat use and increased functioning levels but had no benefit for comorbidity symptoms |
| **Individual-level interventions for any substance use** | | | | | | | |
| Muriungi & Ndetei 2013 [145] | RCT | Effectiveness of psycho-education on depression, hopelessness, suicidality, anxiety and substance use among college students | Intervention group: 1,181 Control group: 1,926 | 4 Psycho education sessions Control: No intervention | Clinical psychologist | Differences in BDI, BHS, BSIS, BAI, ASSIST scores between intervention and control group | Psycho-education was effective in reducing the severity of depression symptoms, hopelessness, suicidality, anxiety and risk of substance abuse at 6 months. |
| **Programs** | | | | | | | |
| Methadone programs | | | | | | | |
| Rhodes 2018 [62] | Qualitative | To evaluate perceptions of persons receiving methadone as regards benefits of the methadone programs | 30 | Methadone programs | - | Perceptions on the recovery potential of methadone programs | Methadone perceived as having recovery potential. |

(*Continued*)

**Table 3.** (Continued)

| Author, Year | Study design | Study objective | Sample size | Name of intervention/ program | Intervention delivered by | Outcomes and measures | Main Findings |
|---|---|---|---|---|---|---|---|
| Rhodes et. al 2015 [61] | Qualitative methods and mathematical modeling | To document the HIV prevention impact of Opioid Substitution Therapy with methadone form the perspective of PWID use | 109 | Opioid substitution therapy with methadone | - | Perceptions of PWID on promise of methadone Projected HIV effects of methadone | Methadone could be an important component of any intervention package aiming to reduce HIV transmission among PWID in Kenya. |
| Needle syringe programs | | | | | | | |
| Ndimbii et al. 2015 [56] | Qualitative | To explore the impact of needle and syringe programs on needle and syringe sharing among PWID use | 109 | Needle and syringe programs | - | Needle and syringe sharing practices before and after needle and syringe programs | Introduction of needle and syringe programs led to significant reductions in needle and syringe sharing. |
| Tobacco cessation programs | | | | | | | |
| Gichuki et al. 2016 [89] | Cross-sectional | To determine the smoking cessation practices of healthcare providers working in public health facilities; training received and barriers to provision of interventions | 400 | Smoking cessation practices | - | Smoking cessation practices; training received; barriers to practice | Practice of smoking cessation interventions was sub-optimal; insufficient training was reported as an important barrier |
| Substance use out-patient programs | | | | | | | |
| Deveau et al. 2010 [84] | Cross-sectional | Evaluate utilization of out-patient addiction services at 4 community-based clinics | 1,847 | Addiction out-patient treatment services | - | Number of clients utilizing services over a 4-year period Abstinence rates | Number of clients participating in treatment services increased from 35 to 479 over the 4-year period 42% reported abstinence from substance use over a 0-36-month period |
| **Population level-interventions for tobacco use** | | | | | | | |
| Perl et al. 2015 [183] | Mixed methods | An assessment of effectiveness and ease of adaptation of anti-tobacco adverts developed in HICs from the perspective of adult smokers and non-smokers | 1078 | Radio and TV anti-tobacco adverts | - | Ratings of effectiveness and ease of adaptation of anti-tobacco ads | Adverts developed in High Income Countries are viable in tobacco control in Africa |
| **Population level-interventions for alcohol use** | | | | | | | |
| Muturi et al. 2016 [55] | Qualitative | To explore community perspectives on alcohol abuse prevention strategies in rural Kenya | 60 | Alcohol abuse prevention strategies | - | Perspectives on alcohol abuse prevention strategies in rural Kenya | Rural communities viewed alcohol abuse prevention interventions as ineffective and messages as unpersuasive in changing this high-risk behavior. |

**Youth and adolescent substance use.**   Three studies focused on substance use among youth and adolescents. In one study, the adolescents perceived that substance use contributed to risky sexual behavior including unprotected sex, transactional sex, and multiple partner sex [58]. The youth identified porn video shows and local brew dens as places where risky sexual encounters between adolescents occurred [59]. Ssewanyana et al. [63] utilized the socio-ecological model to explore perceptions of adolescents and stakeholders on the factors predisposing and contributing to substance use. Substance use among adolescents was perceived to be common and to be due to several socio-cultural factors e.g. access to disposable income, idleness, academic pressure, low self-esteem etc.

**Other topics.**   Utilizing the syndemic theory, one study explored how substance use, violence and HIV risk affect PrEP (Pre-exposure prophylaxis) acceptability, access and intervention needs among male and female sex workers. The study found that co-occurring substance use, and violence experienced by sex workers posed important barriers to PrEP access [41].

A complete description of included qualitative studies is in Table 4.

## Discussion

This is to our knowledge, the first study to summarize empirical work done on substance use and SUDs in Kenya. More than half (77.8%) of the reviewed studies investigated the area of prevalence and risk factors for substance use. Less common were qualitative studies exploring various themes (12.4%) and studies evaluating interventions and programs (9.7%). The first study was conducted in 1982 and since then the number of publications has gradually risen. Most of the research papers (92.4%) were of moderate to high quality. In comparison to two recent scoping reviews conducted in South Africa and Botswana, more research work has been done on substance use in Kenya. Our study found that 185 papers on substance use among Kenyans had been published by the time of the search while Opondo et al. [11] and Tran et al. [10] reported that only 53 and 7 papers focusing on substance use had been published in South Africa (between 1971 and 2017) and in Botswana (between 1983 and 2020) respectively.

### Epidemiology of substance use or SUD

Studies investigating the prevalence, and risk factors for substance use dominated the literature. The studies, which were conducted across a broad range of settings and populations, focused on various substances including alcohol, tobacco, cannabis, opioids, cocaine, sedatives, inhalants, hallucinogens, prescription medication, and ecstasy. In addition, a wide range of important health and socio-demographic factors were examined for their association with substance use. Most studies had robust sample sizes and were conducted using diverse designs including cross-sectional, case-control and cohort. The studies showed a significant burden of substance use among both adults and children and adolescents. In addition, substance use increased the odds of negative mental and physical health outcomes consistent with findings documented in global reports [2, 3]. These findings highlight the importance of making the treatment and prevention for substance use and SUDs of high priority in Kenya.

- Two main evidence gaps were identified within this category: The prevalence and risk factors for substance use among certain vulnerable populations for whom substance use can have severe negative consequences, had not been investigated. For example, no study had included police officers or persons with physical disability, only one study had its participants as pregnant women [113], and only 2 studies had been conducted among HCWs [140, 196].

**Table 4. Studies qualitatively exploring various substance use or SUD related themes.**

| Author, Year | Study objective | Methods of data collection; Study setting & study population | Age and gender distribution | Theoretical frameworks employed | Main findings |
|---|---|---|---|---|---|
| **Injecting drug use and heroin use** | | | | | |
| Yotebieng et al. 2016 [67] | To explore the reproductive health of women of childbearing age who inject drugs and its implications for healthcare | IDIs with 17 women who inject drugs | Age range 20–35 years | Social-ecological theory | Gender inequality and social suffering were reported as driving factors of continued use during pregnancy; healthcare interactions reported as biased toward HIV screening over alcohol and drug screening and education. |
| Beckerleg 2004 [42] | To describe the experiences of injecting heroin users | A combination of anthropology and ethnographic approaches IDIs with 40 persons with injecting heroin use Observation of injecting users in streets and alleys | Age and gender distribution not reported | No theoretical framework mentioned | Heroin injection was perceived as "cool"; Most users were ill-informed on risk of transmission of HIV through injecting practices. |
| Guise et al. 2015 [44] | To explore accounts of transitions from smoking to injecting to understand the role of individual, social and structural processes | The study combined data from two separate studies conducted in Kenya: 1) an in-depth qualitative study of HIV care access for people who inject drugs (n = 118) 2) an ethnographic study of the political economy of the heroin trade in Kenya (n = 92) | Study 1: Age range: 19–49 years; Male 72% Study 2: Age distribution not reported; Male 94% | No theoretical framework mentioned | Transitions from smoking to IDU are experienced as a process of managing a series of resource constraints or of curiosity or search for pleasure. |
| Mburu et al. 2018a [51] | To explore perspectives of women and stakeholders on the intersection between drug use and contraceptive use | IDIs and FGDs with 45 women who inject drugs and 5 stakeholders involved in service provision | Age range 19–56 years Gender distribution of stakeholders not reported | No theoretical framework mentioned | Women linked drug use to amenorrhea hence did not perceive need for contraception |
| Mburu et al. 2018b [47] | to explore the needs and social contexts of women who inject drugs in coastal Kenya | IDIs and FGDs with 45 women who inject drugs and 5 stakeholders involved in service provision | Age range for women & stakeholders 19–56 years PWID 100% female; gender distribution of stakeholders not reported | No theoretical framework mentioned | Several forms of external and self-stigma are experienced by women with IDU. These included internal and external stigma of being a drug user, external gender-related stigma of being a female injecting drug user and external stigma of being HIV positive among participants living with HIV. |
| Mburu et al. 2019a [48] | To document the role of intimate partners in influencing IDU among women | Secondary analysis of a cross sectional qualitative study by Mburu et al 2018 [47] Original study involved IDIs and FGDs with 45 women who inject drugs and 5 stakeholders involved in service provision | Age range for women & stakeholders 19–56 years PWID 100% female; gender distribution of stakeholders not reported | Social-ecological theory | Intimate partners wield significant influence, on the initiation and maintenance of drug use by women; this influence is mediated by inequitable economic and gender-power. |
| Mburu et al. 2020 [49] | To explore factors influencing women's decisions to use drugs during pregnancy | Secondary analysis of a cross sectional qualitative study by Mburu et al 2018 [47] IDIs and FGDs with 45 women who inject drugs and 5 stakeholders involved in service provision | Age range for women & stakeholders 19–56 years PWID 100% female; gender distribution of stakeholders not reported | No theoretical framework mentioned | Women used drugs to cope with stress of unexpected pregnancies, to manage withdrawals. Intimate partners also played roles in facilitating or limiting substance use. |

(*Continued*)

**Table 4.** (Continued)

| Author, Year | Study objective | Methods of data collection; Study setting & study population | Age and gender distribution | Theoretical frameworks employed | Main findings |
|---|---|---|---|---|---|
| Mburu et al. 2019b [50] | To document HIV risks among women who inject drugs in coastal Kenya | Secondary analysis of a cross sectional qualitative study by Mburu et al 2018 [47] IDIs and FGDs with 45 women who inject drugs and 5 stakeholders involved in service provision | Age range for women & stakeholders 19–56 years PWID 100% female; gender distribution of stakeholders not reported | Social-ecological theory | IDU during sex work emerged as an important HIV risk behavior |
| Ndimbii et al. 2018 [57] | To explore utilization of reproductive, maternal, neonatal and child health services among women who inject drugs in coastal Kenya | IDIs and FGDs with 45 women who inject drugs and 5 stakeholders involved in service provision in two coastal towns. | Age range 19–56 years Gender distribution of stakeholders not reported | No theoretical framework mentioned | Drug use interfered with utilization of antenatal and maternal and child health services |
| Syvertsen et al. 2016 [64] | To explore the emergent drug market in Kisumu, western Kenya, from the perspective of PWIDs | Ethnographic methods; 29 IDIs; 151 quantitative surveys with community members reporting IDU | 151 survey participants: mean age 28.8 years; Male 84% Qualitative sample: Mean age 26.7 years; Male 55% | No theoretical framework mentioned | The drug market in Kisumu is dynamic and chaotic reflecting the fluid and adaptive characteristics typical of new drug markets. The drug market is also hidden, erratic, and expensive |
| Mital et al. 2016 [52] | To describe heroin user's experiences during a period of heroin shortage | Rapid assessment methods: 66 KIIs and 15 FGDs with heroin users | At least 18 years of age. Gender distribution not reported | No theoretical framework mentioned | During the shortage, there was desperation and uncertainty, prices for heroin increased, purity decreased, and drug substitution and poly-drug use were practiced. Users transitioned from smoking to injection of heroin during the shortage to compensate for the low quality and quantity. |
| Guise et al. 2019 [45] | To explore experiences of service users on integrated HIV care and methadone treatment | 30 persons on MMT | Mean age: 34 years Male 70% | Material perspective in sociology and Annemarie Mol's analysis of logic of care | Service users preferred that they have choice over whether to seek care for HIV and MMT in a single setting, or separate settings. |
| Syvertsen et al. 2019 [65] | To explore heroin use among FSWs in Kenya to inform services | IDIs with 45 FSWs | Age range: 18–37 years Female 100% | Addiction trajectories concept | Women commonly smoked cocktails containing heroin while using alcohol and other drugs prior to sex work. Most women perceived heroin to boost courage to engage in sex work. Sex work reinforced drug use in ways that both managed and created new risks. |
| Alcohol use | | | | | |
| Ezard et al. 2011 [43] | To describe the burden and pattern of substance use among refugees and IDPs from the perspective of community members and service providers, and identify available resources and interventions for managing the substance use in this population | Rapid assessment and response (RAR) Literature review; 20 Key informant interviews, 14 FGDs (n = 5–12) and 3 group discussions (n-20-34) with substance users; service providers; sex workers; young people; teachers; PLHIV; post-voluntary counselling and testing groups; health workers; pre-formed community groups Direct observation at refugee/IDP sites. | Gender distribution not reported Age range: 17–57 years | No theoretical framework mentioned | Use of alcohol within these populations was widespread and was linked to a range of health and socio-economic problems. Displacement experiences, may make communities vulnerable to substance use and its impact. Access to health services for this population was limited. |

(*Continued*)

**Table 4.** (*Continued*)

| Author, Year | Study objective | Methods of data collection; Study setting & study population | Age and gender distribution | Theoretical frameworks employed | Main findings |
|---|---|---|---|---|---|
| Muturi 2014 [53] | To explore the perceived reproductive health risks associated with alcoholism from the perspective of rural communities in Kenya | Culture-centred approach that emphasizes community engagement in development of interventions; IDIs with 12 opinion leaders and 7 FGDs with 60 community members | Opinion leaders: Age distribution not reported 67% male Community members: Age range 25–57 years 50% male | No theoretical framework mentioned | Heavy alcohol use has severe consequences on sexual and reproductive health |
| Muturi 2015 [54] | To explore rural communities' perspectives on the risk factors for HIV infection among women in alcohol discordant relationships | 60 participants recruited from community-based organizations participated in 7 FGDs | Age range 27–57 years Males 50% | Protection motivation theory | The perceived impact of alcoholism on men's reproductive health and the unmet sexual and reproductive needs of women in alcohol discordant relationships drive women to engage in risky sexual behaviors. |
| Kibicho & Campbell 2019 [46] | To explore the effect of second-generation alcohol consumption on sexual risk behaviors, alcohol misuse, violence and economic stress factors, and HIV infection risk. | 12 FGDs of 80 people from established support groups | At least 18 years of age Male 57.5% | Social-ecological theory and syndemic theory | Second-generation alcohol consumption is prevalent and has profound socio-economic and health effects on households. |
| Velloza et al. 2015 [66] | To describe the stages and processes of change utilized by FSWs participating in an alcohol-reduction intervention | IDIs with 45 FSWs | Age range: 19–48 years Female 100% | Stages of change model | In sessions 1–3, most participants were in the pre-contemplation, contemplation, or preparation stages. In sessions 4–6, most participants were in the action and maintenance stages. In the pre-contemplation stage, participants reported using environmental re-evaluation, consciousness raising, and dramatic relief techniques. In contemplation/ preparation phase, participants said they used self-reevaluation and self-liberation techniques. In action/ maintenance, participants reported using helping relationships, counter-conditioning, reinforcement management, and stimulus control strategies. |
| Othieno et al. 2012 [60] | To explore the factors related to harmful alcohol use and identify interventions aimed at improving adherence to antiretroviral drugs among PLHIV who also use alcohol in a harmful way | Participatory Action Research tools; FGDs with 67 PLHIV and also abusing alcohol and 19 community members drawn from support groups working with PLHIV | Age and gender distribution not reported | No theoretical framework mentioned | Reasons for alcohol use included stigma, to gain social acceptance, to deal with psychological problems, perceived medicinal value, and physical addiction and poverty. Screening and treatment interventions within the community were scarce |

Youth and adolescent substance use

(*Continued*)

**Table 4.** (Continued)

| Author, Year | Study objective | Methods of data collection; Study setting & study population | Age and gender distribution | Theoretical frameworks employed | Main findings |
|---|---|---|---|---|---|
| Njue et al. 2009 [58] | To describe the phenomenon of disco funerals as the setting of risky sexual encounters among youth. | IDIs with 150 adolescents drawn from the community; Observation at 6 disco funerals and 42 places where youth hang-out. | Age range: 15–20 years Male 50% | No theoretical framework mentioned | Drugs and alcohol seemed to facilitate risky unprotected, multiple-partner, coerced, and transactional sex. |
| Njue et al. 2011 [59] | To explore risk situations that can explain the high HIV prevalence among youth in Kisumu town, Kenya | IDIs with 150 adolescents; 4 FGDs and 48 observations at places where youth spend their free time. | Age range: 15–20 years Male 50% | No theoretical framework mentioned | Porn video shows and local brew dens were identified as popular events where unprotected multi-partner, concurrent, coerced and transactional sex occurs between adolescents. |
| Ssewanyana et al. 2018 [63] | To explore perceptions of young people and stakeholders on the types of substances used and the predisposing and protective factors | 11 FGDs with 85 young people (78 adolescents and 7 young adult community representatives); IDIs with 10 stakeholders | Adolescents: aged 10–19 years; 42 males and 36 females Young adult representatives: aged 22–28 years; 3 males and 4 females Stakeholders: Aged 27–51 years; 4 male and 6 females | Social-ecological theory | The use of various substances was common among adolescents. Substance use was due to several interacting social, cultural and community factors e.g. access to disposable income, idleness, academic pressure, low self-esteem, use by close family members etc. |
| **Other topics explored** | | | | | |
| Bazzi et al. 2019 [41] | To explore how substance use, violence and HIV risk shape PrEP acceptability, access and intervention needs among sex workers | 73 Female and male sex workers | Median age (IQR): Female 28 (18 to 42), Male 25 (19 to 41) Male 38.4% | Syndemic theory | Syndemic substance use and violence experienced by sex workers posed important barriers to PrEP access for sex workers. |

- Few studies had explored the epidemiology of hallucinogens, prescription medication, ecstasy, IDU, and emerging substances e.g. synthetic cannabinoids. These substances are a public health threat globally [207, 208] yet their use remains poorly documented in Kenya.

## Interventions and programs

Given the significant documented burden of substance use and SUDs in Kenya, it was surprising that few studies had focused on developing and testing treatment and prevention interventions for SUDs. A possible reason for this is limited expertise in the area of intervention development and testing. For example, research capacity in implementation science has been shown to be limited in resource-poor settings such as ours [209].

Of note is that most of the tested interventions had been delivered by lay providers [40, 90, 175] and primary HCWs [38, 127, 178] indicating a recognition of task-shifting as a strategy for filling the mental health human resource gap in Kenya.

Several research gaps were identified within this category.

- Out of the 11 individual-level interventions tested, nine had targeted harmful alcohol use except one which focused on khat [202] and another that targeted several substances [145]. No studies had evaluated individual-level interventions targeting tobacco and cannabis use, despite the two being the second and third most commonly used substances in Kenya [8]. Further, no individual-level interventions had focused on other important SUDs like opioid, sedative and cocaine use disorders.

- Few studies had evaluated the impact of substance use population-level interventions [55, 183]. Several cost-effective population-level interventions have been recommended by WHO e.g. mass media education and national toll free quit line services for tobacco use, and brief interventions integrated into all levels of primary care for harmful alcohol use [210]. Such strategies need to be tested for scaling up in Kenya.

- None of the interventions had been tested among important vulnerable populations for whom local research already shows a significant burden e.g. children and adolescents, the Lesbian Gay Bisexual Transgender & Queer (LGBTQ) community, HCWs, prisoners, refugees, and IDPs. In addition, no interventions had been tested for police officers and pregnant women, and no studies had evaluated interventions to curb workplace substance use.

- Only one study evaluated digital strategies for delivering substance use interventions [94] yet the feasibility of such strategies has been demonstrated for other mental health disorders in Kenya [211]. Moreover, the time is ripe for adopting such an approach to substance use treatment given the fact that the country currently has a mobile subscriptions penetration of greater than 90% [212].

- No studies had evaluated the impact of other interventions such as mindfulness and physical exercise. Meta-analytic evidence suggests that such strategies hold promise for reducing the frequency and severity of substance use and craving [213, 214].

## Qualitative studies

The qualitative studies focused on a broad range of themes including drivers and impact of substance use, drug markets, patterns of substance use, stigma, and access to treatment. Most of the work however focused on PWID and heroin users. Future qualitative work should explore issues relating to other populations for example persons with other mental disorders, persons with physical disabilities, police officers, and persons using other commonly used substances such as tobacco, khat, and cannabis.

## Limitations

The aim of this systematic review was to provide an overview of the existing literature on substance use and SUD research in Kenya. We therefore did not undertake a meta-analysis and detailed synthesis of the findings of studies included in this review. In addition, variability in measurements of substance use outcomes precluded our ability to more comprehensively summarize the study findings. For quality assessment, detailed assessments using design specific tools were not possible given the diverse methodological approaches utilized in the studies. We therefore used a single tool for the quality assessment of all studies. The results of the quality assessment are therefore to be interpreted with caution. Nonetheless this review describes for the first time the breadth of existing literature on substance use and SUDs in Kenya, identifies research gaps, and provides important directions for future research.

## Conclusion

The purpose of this systematic review was to map the research that has been undertaken on substance use and SUDs in Kenya. Epidemiological studies dominated the literature and indicated a significant burden of substance use among both adults and adolescents. Our findings indicate that there is a dearth of literature regarding interventions for substance use and we are calling for further research in this area. Specifically, interventions ought to be tested not just for alcohol but for other substances as well, and among important at risk populations. In

addition, future research ought to explore the feasibility of delivering substance use interventions using digital means, and the benefit of other interventions such as mindfulness and physical exercise. Future qualitative work should aim at providing in-depth perspectives on substance use among populations excluded from existing literature e.g. police officers, persons using other substances such as tobacco, cannabis and khat, and persons with physical disability.

## Supporting information

**S1 Checklist. PRISMA checklist.**
(DOCX)

**S1 File. Search terms for PsychINFO.**
(PDF)

## Author Contributions

**Conceptualization:** Florence Jaguga, Sarah Kanana Kiburi.

**Formal analysis:** Florence Jaguga, Eunice Temet, Julius Barasa, Serah Karanja, Lizz Kinyua, Edith Kamaru Kwobah.

**Methodology:** Florence Jaguga, Sarah Kanana Kiburi.

**Supervision:** Sarah Kanana Kiburi, Edith Kamaru Kwobah.

**Validation:** Florence Jaguga, Eunice Temet, Julius Barasa.

**Writing – original draft:** Florence Jaguga.

**Writing – review & editing:** Florence Jaguga, Sarah Kanana Kiburi, Eunice Temet, Julius Barasa, Serah Karanja, Lizz Kinyua, Edith Kamaru Kwobah.

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
