## [Decision Letter · Decision Letter 0]

28 Mar 2022

PONE-D-22-00681A systematic review of
substance use and substance use disorder research in
KenyaPLOS ONE

Dear Dr. JAGUGA,

Thank you for submitting your manuscript to PLOS ONE. After careful consideration, we
feel that it has merit but does not fully meet PLOS ONE’s publication criteria as it
currently stands. Therefore, we invite you to submit a revised version of the
manuscript that addresses the points raised during the review
process.

Please submit your revised manuscript by May 12 2022 11:59PM. If you will need more
time than this to complete your revisions, please reply to this message or contact
the journal office at plosone@plos.org. When
you're ready to submit your revision, log on to https://www.editorialmanager.com/pone/ and select the 'Submissions
Needing Revision' folder to locate your manuscript file.

Please include the following items when submitting your revised
manuscript:A rebuttal letter that responds to each point raised by the academic
editor and reviewer(s). You should upload this letter as a separate file
labeled 'Response to Reviewers'.A marked-up copy of your manuscript that highlights changes made to the
original version. You should upload this as a separate file labeled
'Revised Manuscript with Track Changes'.An unmarked version of your revised paper without tracked changes. You
should upload this as a separate file labeled 'Manuscript'.

If you would like to make changes to your financial disclosure, please include your
updated statement in your cover letter. Guidelines for resubmitting your figure
files are available below the reviewer comments at the end of this letter.

We look forward to receiving your revised manuscript.

Kind regards,

Judith I Tsui

Academic Editor

PLOS ONE

Journal Requirements:

- Tsuei, S.HT., Clair, V., Mutiso, V. et al. Factors Influencing Lay and Professional
Health Workers’ Self-efficacy in Identification and Intervention for Alcohol,
Tobacco, and Other Substance Use Disorders in Kenya. Int J Ment Health Addiction 15,
766–781 (2017). https://doi.org/10.1007/s11469-017-9775-6

The text that needs to be addressed involves lines 279-283 in your submission.

In your revision ensure you cite all your sources (including your own works), and
quote or rephrase any duplicated text outside the methods section. Further
consideration is dependent on these concerns being addressed

Reviewers' comments:

Reviewer's Responses to Questions

**Comments to the Author**

1. Is the manuscript technically sound, and do the data support the conclusions?

Reviewer #1: Yes

Reviewer #2: Partly

2. Has the statistical analysis been performed
appropriately and rigorously? 

Reviewer #1: N/A

Reviewer #2: N/A

3. Have the authors made all data underlying the
findings in their manuscript fully available?

Reviewer #1: Yes

Reviewer #2: Yes

4. Is the manuscript presented in an intelligible
fashion and written in standard English?

Reviewer #1: No

Reviewer #2: Yes

5. Review Comments to the Author

Reviewer #1: This manuscript adds value in an under-research area by summarizing main
learnings from substance use and substance use disorder research in Kenya. Major
revisions are needed though for the article to be presented as a scientifically-
acceptable piece. These revision include punctuation, grammar errors (eg:
inappropriate use of upper case letter on line 84- 86), and overall flow of some
sentences such as line 31, line 39, line 50, line 58, line 69, line 83, line 87- 91,
line 104, line 172, line 175 to list a few.

In addition to these revision, below are proposed consideration:

- In the abstract, please specify the start date used in the search strategy.

- Line 53- tobacco kills 8million people where? Worldwide? On a specific continent?
Please specify

-Line 57- you mentioned one consequence so far ie death which others are you
referring to here?

- Line 104- inception of what?

-Line 111 who checked the duplicates? Was the software used for this or did the
authors do it? It's a bit unclear

-Line 124-125: Were mixed methods studies included as well? The way this is phrased
it sounds like "all designs" refers more to qualitative and quantitative studies

-Avoid over using "/" in sentences. If need be list item a "or" b throughout the
manuscript. Eg: substance use or SUDs

-Line 182- 183- did you mean that 13 additional studies were identified? Please
consider reviewing and rephrasing your sentences to improve clarity

-Line 185- Is "These" referring to the studies? If yes, can you be a bit more
explicit?

-Line 238- are you referring to MSM who are commercial sex workers? Please use
appropriate languages throughout the manuscript

- For the result presentation, it might be helpful to have as part of the main
manuscript (not supplemental information) a summary table of the final literature
reviewed including information on the title of the article, authors, methods,
findings and gap from the articles that were included in the review instead of
having long references throughout the result section.

- Line 313 Lay healthcare providers might be more appropriate same for line 314 for
primary healthcare workers not primary care workers

-Line 331-332 that last sentence seems incomplete, please consider reviewing it

-Line 371- 372- What are estimates then on what has been done elsewhere in SSA? Is
this conclusion based mainly on the 2 scoping work from SA and Bostwana? How about
other SSA countries including countries neighboring Kenya like Uganda, Tanzania,
etc?

How do you define a lot?

-Line 392- Emerging substances like which ones?

- Line 404- Was the study specifically assessing feasibility? If that was not the
case, making such claim is misleading

-

Reviewer #2: This systematic review highlights several gaps in licit and illicit
substance use (SU) and substance use disorder (SUD) literature within Kenya, with
the goal of summarizing research within three broad domains: (1) epidemiologic
studies, (2) intervention and/or programs and (3) qualitative studies. The authors
apply sound methods, with attention to details around decision-making processes when
including articles in their review. The attention to target study populations (e.g.,
community, hospitals, prisons, etc.) is extremely valuable and calls for additional
studies within specific populations. In addition, the authors make the case that
their review is needed in order to address Kenya’s Vision 2030 and moves towards
accomplishing SDG’s. I commend the authors for completing this large undertaking and
offer feedback to strengthen and improve their paper.

Major Edits

• There is an absolute need for SU and SUD systematic review; however, this paper may
have limited applications in its current state. In the introduction, the authors
state this paper will “guide future research efforts”; however, most SUD researchers
work with one substance or one category of substances. It would be helpful within
the key findings sections to expand on SU categories, which are discussed briefly in
the introduction (e.g., tobacco, alcohol, opioids, cannabis, and stimulants.)
Another option may be to reformat the paragraphs according to SU categories and
discuss the current epidemiologic, interventions/programs, and qualitative
studies.

• In your criteria, you do not mention whether you included studies conducted out of
methadone clinics or harm reduction sites (i.e., drop-in centres, NSPs),
specifically. However, when I look over the publications, several were conducted
within these sites. Please clarify whether these terms were part of your search
categories and include them on Page 11, lines 215-217.

• Throughout the descriptions and key findings sections, there should be more
syntheses of the data instead of frequencies, which are already conveyed in your
tables. For example, under the epidemiology section of SU/SUD, you say that 47% of
the studies used evidence-based diagnostic tools, but this should be followed by the
key findings of those studies (i.e., X-X% of participants indicated hazardous or
harmful alcohol consumption, and X-X% of participants indicated alcohol dependence.)
This is just one example, but all of the key finding’s sections should provide more
data syntheses.

• As it stands, the key findings and other findings sections are a little difficult
to follow and are heavily focused on alcohol and tobacco use. For example, in the
epidemiologic key findings section the paragraphs are organized as follows: (1)
youth and substance use, (2) adults and tobacco use, (3) adults and alcohol use, and
(4) two case control studies. Again, this may have a better flow if the authors
organized the key findings by SU categories (e.g., tobacco, alcohol, opioids,
cannabis, and stimulants.) By structuring the paragraphs by SU categories, the
reader is able to quickly decipher where there are gaps in the literature.
Alternatively, the authors may want to consider narrowing the scope of their paper
by solely focusing on alcohol and tobacco use, which seem to be the main focus
throughout the paper.

• In the qualitative study key findings section, most of the studies apply frameworks
and/or theories to their analysis (e.g., stages of change, risk environment
framework), which should be synthesized and included as a column in Additional File
5/Qualitative Studies.

Minor Edits

• Please review the PLOS ONE Guidelines on formatting references and edit
references.

• Page 11 (line 220) “People with injecting drug use” should be “people (or persons)
who inject drugs.”

• Page 11 (line 221) “Men who have Sex with Men” should not contain capital
letters.

• Page 11 (lines 218-225) This section does not sum up to the total studies in the
epidemiology section n=144.

• Page 11 (line 210-213) Please be consistent in how you mention the study designs
with corresponding references. This was completed in the interventions and programs
section, but not for the epidemiological studies.

• Page 15 (lines 299-303) Conversely, please indicate in the programs and
intervention section, how may studies were included in each of the study
designs.

• Page 12 (line 229) typo, please change to “opioids (n=21)”

• In the findings section, please define “hospital,” and whether this includes
methadone clinics.

• Page 20 (line 398) “Substance use” should be “substance use disorder.”

• Page 21 (line 423-424) “Mental disorders” should be “mental health disorders.”

• Additional File 3/Epidemiological Studies: The SU category should not include how
people consume their drugs (“injection drugs”), which is only seen a few times, but
what drugs categories were examined. Please be more specific than “illicit
drugs.”

• Additional File 4/Interventions and Program: Please review the sample sizes for
each study, particularly for those with “not reported.”

6. PLOS authors have the option to publish the peer
review history of their article (what does this mean?). If published, this will
include your full peer review and any attached files.

If you choose “no”, your identity will remain anonymous but your review may still be
made public.

**Do you want your identity to be public for this peer review?** For
information about this choice, including consent withdrawal, please see our
Privacy Policy.

Reviewer #1: No

Reviewer #2: No

---

## [Author Response · Author response to Decision Letter 0]

12 May 2022

Reviewer #1: This manuscript adds value in an under-research area by summarizing main
learnings from substance use and substance use disorder research in Kenya. Major
revisions are needed though for the article to be presented as a scientifically-
acceptable piece. These revision include punctuation, grammar errors (eg:
inappropriate use of upper case letter on line 84- 86), and overall flow of some
sentences such as line 31, line 39, line 50, line 58, line 69, line 83, line 87- 91,
line 104, line 172, line 175 to list a few. 

We thank the reviewer for this comment. We have thoroughly proof read the paper and
made corrections to grammar and punctuation.

- In the abstract, please specify the start date used in the search strategy. 

We have specified that the search was conducted from inception (line 27).

- Line 53- tobacco kills 8million people where? Worldwide? On a specific continent?
Please specify 

We have clarified that it is worldwide (line 58)

-Line 57- you mentioned one consequence so far ie death which others are you
referring to here?

The paragraph has been revised to include health consequences of alcohol, tobacco and
other substances (line 58-63)

- Line 104- inception of what? 

Inception means from the earliest available study. This term is commonly used in
systematic review searches when no date limits have been set

-Line 111 who checked the duplicates? Was the software used for this or did the
authors do it? It's a bit unclear

The Mendeley Reference manager was used to identify and remove duplicates. This has
been clarified on line 116-117.

-Line 124-125: Were mixed methods studies included as well? The way this is phrased
it sounds like "all designs" refers more to qualitative and quantitative studies

Yes, we included studies with qualitative, quantitative and mixed methods designs.
This has now been clarified (line 133).

-Avoid over using "/" in sentences. If need be list item a "or" b throughout the
manuscript. Eg: substance use or SUDs

This has been corrected throughout the manuscript

-Line 182- 183- did you mean that 13 additional studies were identified? Please
consider reviewing and rephrasing your sentences to improve clarity

The sentence has been reviewed to improve clarity (line 208)

-Line 185- Is "These" referring to the studies? If yes, can you be a bit more
explicit?

We have reworded the sentence to make it more explicit (line 210)

-Line 238- are you referring to MSM who are commercial sex workers? Please use
appropriate languages throughout the manuscript

The authors are referring to MSM who were commercial sex workers. We have corrected
this (line 273).

- For the result presentation, it might be helpful to have as part of the main
manuscript (not supplemental information) a summary table of the final literature
reviewed including information on the title of the article, authors, methods,
findings and gap from the articles that were included in the review instead of
having long references throughout the result section.

We have included the tables within the main manuscript (line 367, 439, 504)

- Line 313 Lay healthcare providers might be more appropriate same for line 314 for
primary healthcare workers not primary care workers 

This has been corrected line 388, 391, 395, 551, 552

-Line 331-332 that last sentence seems incomplete, please consider reviewing it

This sentence has been revised (line 428-430)

-Line 371- 372- What are estimates then on what has been done elsewhere in SSA? Is
this conclusion based mainly on the 2 scoping work from SA and Bostwana? How about
other SSA countries including countries neighboring Kenya like Uganda, Tanzania,
etc?

We have reworded the paragraph to show that we are comparing our findings with
available scoping reviews (line 513-520)

How do you define a lot? We have revised this sentence and used the word “more…”
(line 515)

-Line 392- Emerging substances like which ones?

An example has been given (line 541)

- Line 404- Was the study specifically assessing feasibility? If that was not the
case, making such claim is misleading

This line has been deleted (line 554).

-

Reviewer #2: This systematic review highlights several gaps in licit and illicit
substance use (SU) and substance use disorder (SUD) literature within Kenya, with
the goal of summarizing research within three broad domains: (1) epidemiologic
studies, (2) intervention and/or programs and (3) qualitative studies. The authors
apply sound methods, with attention to details around decision-making processes when
including articles in their review. The attention to target study populations (e.g.,
community, hospitals, prisons, etc.) is extremely valuable and calls for additional
studies within specific populations. In addition, the authors make the case that
their review is needed in order to address Kenya’s Vision 2030 and moves towards
accomplishing SDG’s. I commend the authors for completing this large undertaking and
offer feedback to strengthen and improve their paper.

We thank the reviewer for their comments.

Major Edits

• There is an absolute need for SU and SUD systematic review; however, this paper may
have limited applications in its current state. In the introduction, the authors
state this paper will “guide future research efforts”; however, most SUD researchers
work with one substance or one category of substances. It would be helpful within
the key findings sections to expand on SU categories, which are discussed briefly in
the introduction (e.g., tobacco, alcohol, opioids, cannabis, and stimulants.)
Another option may be to reformat the paragraphs according to SU categories and
discuss the current epidemiologic, interventions/programs, and qualitative
studies.

We acknowledge this comment. We have organized the key findings sections by substance
use categories and expanded on the findings (line 162, 266-366, 373-439,
446-503).

• In your criteria, you do not mention whether you included studies conducted out of
methadone clinics or harm reduction sites (i.e., drop-in centres, NSPs),
specifically. However, when I look over the publications, several were conducted
within these sites. Please clarify whether these terms were part of your search
categories and include them on Page 11, lines 215-217.

NSP sites has been included in the general characteristics of epidemiological studies
(line 248)

• Throughout the descriptions and key findings sections, there should be more
syntheses of the data instead of frequencies, which are already conveyed in your
tables. For example, under the epidemiology section of SU/SUD, you say that 47% of
the studies used evidence-based diagnostic tools, but this should be followed by the
key findings of those studies (i.e., X-X% of participants indicated hazardous or
harmful alcohol consumption, and X-X% of participants indicated alcohol dependence.)
This is just one example, but all of the key finding’s sections should provide more
data syntheses.

• We have now provided more synthesis of data in the results section

(line 266-366, 373-439, 446-503).

• As it stands, the key findings and other findings sections are a little difficult
to follow and are heavily focused on alcohol and tobacco use. For example, in the
epidemiologic key findings section the paragraphs are organized as follows: (1)
youth and substance use, (2) adults and tobacco use, (3) adults and alcohol use, and
(4) two case control studies. Again, this may have a better flow if the authors
organized the key findings by SU categories (e.g., tobacco, alcohol, opioids,
cannabis, and stimulants.) By structuring the paragraphs by SU categories, the
reader is able to quickly decipher where there are gaps in the literature.
Alternatively, the authors may want to consider narrowing the scope of their paper
by solely focusing on alcohol and tobacco use, which seem to be the main focus
throughout the paper.

We acknowledge this comment. We have organized the key findings sections by substance
use categories and expanded on the findings (line 162, 266-366, 373-439,
446-503).

• In the qualitative study key findings section, most of the studies apply frameworks
and/or theories to their analysis (e.g., stages of change, risk environment
framework), which should be synthesized and included as a column in Additional File
5/Qualitative Studies.

We have incorporated the theoretical frameworks into the results section (line
468,478, 492, 499), and added a column presenting information on theoretical
frameworks to the table 4 (line 505).

Minor Edits

• Please review the PLOS ONE Guidelines on formatting references and edit
references.

The references have been edited in line with PLOS one guidelines

• Page 11 (line 220) “People with injecting drug use” should be “people (or persons)
who inject drugs.”

This has been corrected (line 251)

• Page 11 (line 221) “Men who have Sex with Men” should not contain capital
letters.

This has been corrected (line 252)

• Page 11 (lines 218-225) This section does not sum up to the total studies in the
epidemiology section n=144.

Yes. This is true because some populations overlapped e.g. some studies were
conducted among general population adults with NCDs.

• Page 11 (line 210-213) Please be consistent in how you mention the study designs
with corresponding references. This was completed in the interventions and programs
section, but not for the epidemiological studies.

We have now deleted references in the general description section for the
intervention studies (line 375-378) and qualitative studies (448-457) to ensure
uniformity

• Page 15 (lines 299-303) Conversely, please indicate in the programs and
intervention section, how may studies were included in each of the study
designs.

This has been indicated. Line 386-389

• Page 12 (line 229) typo, please change to “opioids (n=21)”

This has been corrected. Line 261

• In the findings section, please define “hospital,” and whether this includes
methadone clinics.

We have separated out studies done within hospitals and those done within methadone
clinics (line 247; Kisilu et al. 2019 on table 2 line 367)

• Page 20 (line 398) “Substance use” should be “substance use disorder.”

This has been corrected. Line 551

• Page 21 (line 423-424) “Mental disorders” should be “mental health disorders.”

This has been corrected. Line 578

• Additional File 3/Epidemiological Studies: The SU category should not include how
people consume their drugs (“injection drugs”), which is only seen a few times, but
what drugs categories were examined. Please be more specific than “illicit
drugs.”

The studies described the substances as just IDU and illicit substances, and did not
provide descriptions of the specific substances assessed for. We have included the
phrase ‘not specified’ next to the term illicit drugs and IDU for clarity. (Table 2
line 367)

• Additional File 4/Interventions and Program: Please review the sample sizes for
each study, particularly for those with “not reported.”

These were reviewed and appropriate sample sizes reported (table 3 line 443)

Editors’ comments

- Tsuei, S.HT., Clair, V., Mutiso, V. et al. Factors Influencing Lay and Professional
Health Workers’ Self-efficacy in Identification and Intervention for Alcohol,
Tobacco, and Other Substance Use Disorders in Kenya. Int J Ment Health Addiction 15,
766–781 (2017). https://doi.org/10.1007/s11469-017-9775-6

The text that needs to be addressed involves lines 279-283 in your submission.

We have addressed this (line 349-354)

About data availability. All analyzed data has been included in the main manuscript
and in the supporting information files 1 and 2. (line 1264)

---

## [Editor Report · Decision Letter 1]

19 May 2022

A systematic review of substance use and substance use disorder research in Kenya

PONE-D-22-00681R1

Dear Dr. JAGUGA,

We’re pleased to inform you that your manuscript has been judged scientifically
suitable for publication and will be formally accepted for publication once it meets
all outstanding technical requirements.

Kind regards,

Judith I Tsui

Academic Editor

PLOS ONE
---

## [Editor Report · Acceptance letter]

26 May 2022

PONE-D-22-00681R1 

A systematic review of substance use and substance use disorder research in Kenya 

Dear Dr. Jaguga:

I'm pleased to inform you that your manuscript has been deemed suitable for
publication in PLOS ONE. Congratulations! Your manuscript is now with our production
department. 

Kind regards, 

on behalf of

Dr. Judith I Tsui 

Academic Editor

PLOS ONE